# Adaptive biasing of action-selective cortical build-up activity by stimulus history

Anke Braun[1,2,3]*, Tobias H Donner[1,4]*

[1]Section Computational Cognitive Neuroscience, Department of Neurophysiology and Pathophysiology, University Medical Center Hamburg-Eppendorf, Hamburg, Germany; [2]Charité – Universitätsmedizin Berlin, corporate member of Freie Universität Berlin and Humboldt-Universität zu Berlin, Department of Psychiatry and Neurosciences, Berlin, Germany; [3]Charité – Universitätsmedizin Berlin, corporate member of Freie Universität Berlin and Humboldt-Universität zu Berlin, Department of Child and Adolescent Psychiatry, Berlin, Germany; [4]Bernstein Center for Computational Neuroscience, Charité – Universitätsmedizin Berlin, Berlin, Germany

*For correspondence:
anke.braun86@gmail.com (AB);
t.donner@uke.de (THD)

Competing interest: The authors declare that no competing interests exist.

**Abstract** Decisions under uncertainty are often biased by the history of preceding sensory input, behavioral choices, or received outcomes. Behavioral studies of perceptual decisions suggest that such history-dependent biases affect the accumulation of evidence and can be adapted to the correlation structure of the sensory environment. Here, we systematically varied this correlation structure while human participants performed a canonical perceptual choice task. We tracked the trial-by-trial variations of history biases via behavioral modeling and of a neural signature of decision formation via magnetoencephalography (MEG). The history bias was flexibly adapted to the environment and exerted a selective effect on the build-up (not baseline level) of action-selective motor cortical activity during decision formation. This effect added to the impact of the current stimulus. We conclude that the build-up of action plans in human motor cortical circuits is shaped by dynamic prior expectations that result from an adaptive interaction with the environment.

## eLife assessment

In uncertain conditions, decisions are not made in isolation but are rather biased by the recent past. This new work provides **valuable** insights into these history biases in human perceptual decision-making, by characterizing the neural correlates of stimulus history biases and their short-term dynamics. The study provides **compelling** behavioral and MEG evidence that humans adapt their history biases to the correlation structure of uncertain sensory environments.

## Introduction

Perceptual decisions made in the face of uncertain sensory evidence are often biased by previous stimuli, choices, and choice outcomes (*Gold et al., 2008*; *Busse et al., 2011*; *de Lange et al., 2013*; *Akaishi et al., 2014*; *Fischer and Whitney, 2014*; *Fründ et al., 2014*; *Abrahamyan et al., 2016*; *Pape and Siegel, 2016*; *St John-Saaltink et al., 2016*; *Fritsche et al., 2017*; *Hwang et al., 2017*; *Urai et al., 2017*; *Braun et al., 2018*). In most standard laboratory tasks, the environmental state (i.e., stimulus category) is uncorrelated across trials. In that context, such history biases tend to impair performance (*Abrahamyan et al., 2016*). However, when the environmental state exhibits some stability across trials, as is common for natural environments (*Yu and Cohen, 2009*; *Glaze et al., 2015*), trial

history biases substantially improve performance (*Braun et al., 2018*). Previous behavioral work shows that humans and other animals flexibly adapt their trial history bias to the correlation structure of the environment (*Abrahamyan et al., 2016*; *Kim et al., 2017*; *Braun et al., 2018*; *Hermoso-Mendizabal et al., 2020*).

How do adaptive history biases influence the formation of subsequent decisions? Prominent models conceptualize the decision formation as the accumulation of noisy sensory evidence into a decision variable (DV) that grows with time until a bound for one of the choice alternatives is crossed and the corresponding action is initiated (*Bogacz et al., 2006*; *Gold and Shadlen, 2007*; *Ratcliff and McKoon, 2008*; *Brunton et al., 2013*; *Ossmy et al., 2013*). In such accumulator models, history biases may shift the starting point of the DV before evidence onset and/or bias the evidence accumulation per se. Behavioral modeling indicates that individual differences in the idiosyncratic history biases occurring in random environments are better explained by biases of evidence accumulation than by starting point biases (*Urai et al., 2019*). Such effects have neither been assessed for adaptive biases in structured (stable or systematically alternating) environments nor have they been unraveled at the neural level. We hypothesized that adaptive history biases translate into a biased build-up rate (accumulation bias) of neural signatures of the DV, more so than an offset before decision formation (starting point), in a fashion that depends on the correlation structure of the environment.

Neural signals exhibiting functional properties of the DV have been observed in parietal and frontal cortical areas involved in action planning in both primates (*Shadlen and Kiani, 2013*; *Peixoto et al., 2021*) and rodents (*Hanks et al., 2015*; *Brody and Hanks, 2016*). Specifically, when choices are reported with hand movements, hallmark signatures of the DV are evident in motor preparatory activity in primate (human and monkey) premotor and primary motor (M1) cortex. In human motor cortex, this selective motor preparatory activity is expressed in a suppression of ongoing beta-band oscillations contralateral to the upcoming hand movement, accompanied by an enhancement of gamma-band power (*Crone et al., 1998a*; *Crone et al., 1998b*; *Donner et al., 2009*) and likely spiking activity. While the origin of this beta-power suppression remains under study (*Sherman et al., 2016*; *Little et al., 2019*) we here use it as a functional marker of the DV encoded in local patters of spiking activity: Like this spiking activity (*Shadlen and Kiani, 2013*; *Peixoto et al., 2021*), the beta-band suppression (i) encodes the specific choice that will later be reported, (ii) gradually builds up during decision formation, with a rate that scales with evidence strength, and (iii), in reaction time tasks, converges on a common level just before action execution (*Donner et al., 2009*; *O'Connell et al., 2012*; *Wyart et al., 2012*; *de Lange et al., 2013*; *Fischer et al., 2018*; *Wilming et al., 2020*; *Murphy et al., 2021*).

We combined a canonical decision-making task, discrimination of the net motion direction of dynamic random dot patterns (*Gold and Shadlen, 2007*), with a systematic manipulation of the environmental correlation structure. Our manipulation was motivated from an ecological perspective (*Mobbs et al., 2018*), specifically, the insights that (i) natural environments are commonly structured and (ii) history biases that change flexibly with this environmental structure are a hallmark of adaptive behavior. Because previous work on such correlated environments was purely behavioral (*Abrahamyan et al., 2016*; *Kim et al., 2017*; *Braun et al., 2018*; *Hermoso-Mendizabal et al., 2020*), the neural signatures of adaptive history biases have remained unknown. Although several previous studies have identified neural correlates of history biases in standard perceptual choice tasks (i.e., using unstructured environments) (*Talluri et al., 2021*), all but one study performed in monkeys (*Mochol et al., 2021*) focused on static representations of the bias in ongoing activity preceding the new decision. Therefore, it has remained unknown whether such a dynamic bias during evidence accumulation exists in the human brain.

Single-trial behavioral modeling uncovered the resulting history-dependent biases as well as their flexible adjustment to the environmental correlation structure. Relating the model-inferred time-varying history bias to magnetoencephalography (MEG) measurements of the pre-trial baseline state and subsequent build-up rate of action-selective motor cortical population activity identified a neural signature of this adaptive bias in the latter, not the former. In sum, we show that the sign and rate of the build-up of a selective neural marker of DV during evidence accumulation track a dynamic history bias that is adapted to the environmental structure.

# Results

Human participants (*N*=38) performed a random dot motion (up vs. down) discrimination task with varying levels of motion strength spanning psychophysical threshold (*Figure 1A*, Materials and methods). We alternated the task, in pseudo-random order, between three different sensory environments with distinct repetition probabilities of stimulus categories (i.e., motion directions) across trials, referred to as neutral, repetitive, and alternating, respectively (*Figure 1B*). These three environments were characterized by approximately equal fractions of upward and downward motion stimuli, and they were presented in blocks of 99 trials each, separated by pauses (Materials and methods). Participants were not informed about the existence of these different environments and received outcome feedback after each choice.

## Adjustment of history biases to environmental context

We expected that the history biases would vary systematically between these different sensory environments, as observed in previous work (*Abrahamyan et al., 2016*; *Braun et al., 2018*; *Hermoso-Mendizabal et al., 2020*). Because the feedback after each trial disambiguated the previous stimulus category, we further expected that subjects might use that information for adjusting their history biases to the environment. We observed an indication of such an adjustment in their psychometric functions, when those were fit conditioned on the previous stimulus category (*Figure 1C*). In all three environments, previous category-dependent psychometric functions were shifted horizontally, indicative of a history bias (repetitive: $t=8.133$, $p<10^{-4}$, neutral: $t=4.218$, $p=0.0002$, alternating: $t=-2.276$, $p=0.0287$; two-tailed t tests). Critically, these shifts pointed in opposite directions for the two structured environments, with a strong tendency to repeat the previous category in repetitive and a tendency to alternate the previous category in alternating, which highlights the adaptive nature of the history biases (*Figure 1C*). The previous stimulus category had no effect on perceptual sensitivity (history-dependent psychometric slopes: repetitive: $t=-0.0397$, $p=0.969$, $Bf_{10}=0.175$, neutral: $t=-0.623$, $p=0.537$, $Bf_{10}=0.209$, alternating: $t=0.094$, $p=0.926$, $Bf_{10}=0.175$; two-tailed t tests and Bayes factors).

We used a statistical model to quantify participants' history biases in a more comprehensive fashion and estimate single-trial bias time courses for the interrogation of MEG data in the subsequent sections. The model was fit separately to the choice behavior from each sensory environment and captured the history bias as a linear combination of the choices and stimulus categories from the recent trials (Materials and methods). We used a cross-validation procedure to select the best fitting model order (i.e., number of previous trials contributing to the bias), separately for each individual and each environment (*Figure 1—figure supplement 1*) and applied this model to independent data in order to estimate subjects' history weights (*Figure 1D*; *Figure 1—figure supplement 2*) and construct bias time courses (*Figure 1E*). The analyses presented in the following included only those subjects (all but two), which showed a best fitting lag larger than 0 in at least one of the two biased environments (commonly repetitive, *Figure 1—figure supplement 1*).

The estimated model parameters (cross-validated regression weights, *Figure 1D*, *Figure 1—figure supplement 2*) showed a pattern in line with the psychometric function shifts in *Figure 1C*. In *Figure 1D*, positive regression weights for the previous stimulus category indicated a tendency for subjects to repeat (in their choice) the previously shown stimulus category. Likewise, negative weights indicated a tendency to alternate the choice relative to the previous stimulus category. The impact of the previous trial stimulus category on current choice was different from zero in all three environments, including neutral (*Figure 1D*; repetitive: $p<10^{-5}$; alternating: $p=0.0003$; neutral: $p=0.0002$; two-tailed permutation tests). But critically, in both biased environments, this impact was different from neutral and shifted in opposite directions, indicating a tendency to repeat the previous stimulus category in repetitive and vice versa in alternating (*Figure 1D*; repetitive vs. neutral: $p<10^{-5}$; alternating vs. neutral: $p<10^{-5}$; two-tailed permutation tests).

The impact of the previous choice on the current choice tended to be overall weaker, more idiosyncratic, and less systematically related to the sensory environment than the impact of the previous stimulus (*Figure 1—figure supplement 2A*). Indeed, the shift in stimulus weights between each biased condition and neutral was significantly larger than the corresponding shift in choice weights (repetitive $p=0.0002$, alternating: $p=0.0026$; two-tailed permutation test). There was little contribution of the stimulus categories from trials further back in time (*Figure 1—figure supplement 2B*). Overall,

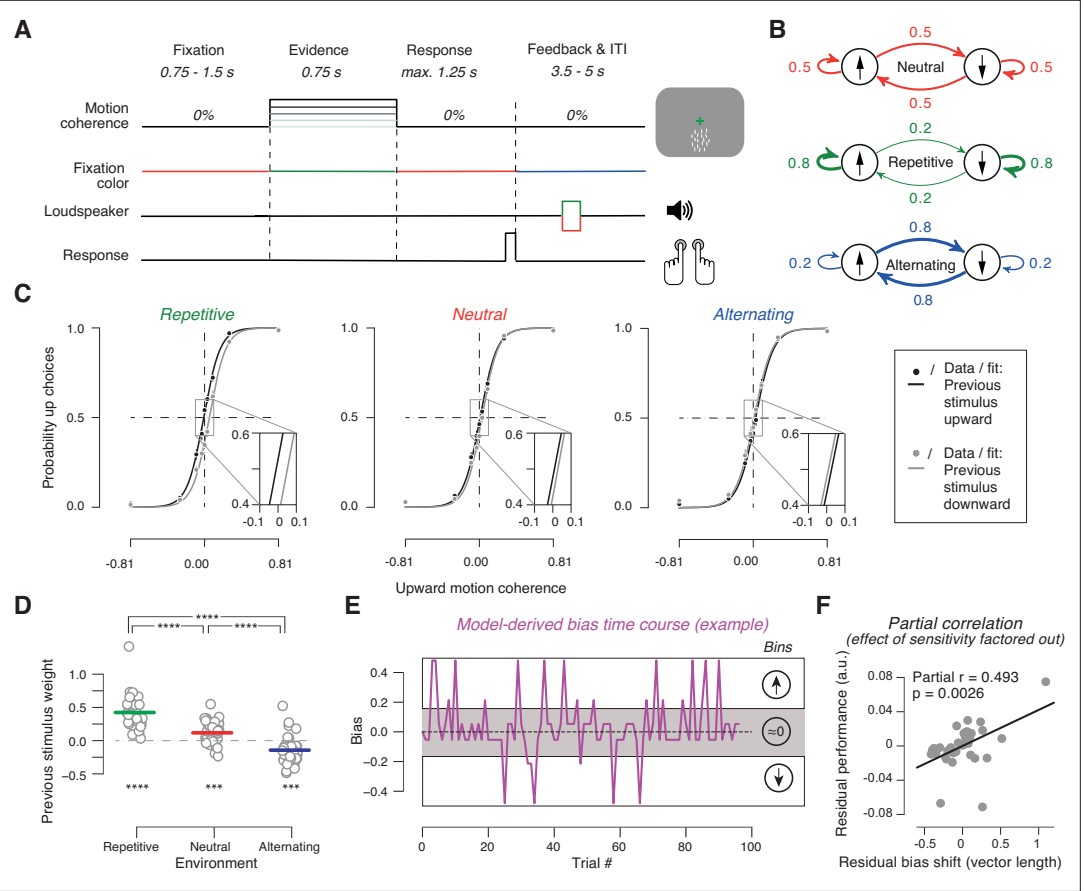

**Figure 1.** Task and behavior in the different sensory environments. (**A**) Time course of events during a trial. Participants judged the net direction of motion of random dot kinematograms with varying levels of motion coherence and direction. 0% coherent motion was presented throughout the trial. Color switch of fixation cross indicated the onset of the decision interval with coherent motion (or 0% coherence on some trials). After 0.75 s, the color of the fixation cross switched back to red, to prompt the choice. After the button-press or 1.25 s deadline, the fixation cross turned blue indicating the variable inter-trial interval with auditory feedback. (**B**) Manipulation of stimulus environments through variation of repetition probability of motion direction across trials. Repetition probability was 0.8 (repetitive), 0.5 (neutral), or 0.2 (alternating). Adapted from *Braun et al., 2018*, B; Creative Commons Attribution License Creative Commons Attribution 4.0 International. The copyright holder has granted permissions to publish under CC BY 4.0 licence. (**C**) Psychometric functions conditioned on previous stimulus category (group average), for the three environments (n = 38). Vertical lines, SEM (most are smaller than data points); insets, close-ups of the part in rectangle around 0% coherence indicating the systematic shift of history bias between the environments. (**D**) Impact of previous stimulus categories on current choice for lag 1. Circles refer to values from individual participants. Lines refer to group means. ***p<0.001, ****p<0.0001 two-tailed permutation test. (**E**) Single-trial history bias estimates for an example participant and block from the neutral environment. Positive values correspond to a bias for choice 'up' and negative values correspond to a bias for choice 'down'. The magnitude indicates the strength of the bias. When binned into three bins of equal size, the low bin contains trials with a bias for choice 'down', the medium bin contains trials with a bias around zero, and the high bin contains trials with a bias for choice 'up'. (**F**) Bias adjustment improves performance. Partial regression (Pearson correlation) between length of the vector of previous choice weights plotted against previous stimulus weights between repetitive and alternating in *Figure 1—figure supplement 2A* and proportion of correct choices averaged across repetitive and alternating while factoring out the effect of sensitivity. Data points are the residuals from two separate regressions: length of vector difference on sensitivity (x-axis) and sensitivity on proportion correct (y-axis).

The online version of this article includes the following figure supplement(s) for figure 1:

**Figure supplement 1.** Best fitting model orders for behavioral history bias.

**Figure supplement 2.** Patterns of individual stimulus history biases across environments.

**Figure supplement 3.** Performance in biased environments depends on strength of previous stimulus weights.

the pattern of model parameters is consistent with our expectation that participants' adjustment of their history biases would be governed by the previous stimulus category, which was disambiguated through the trial-by-trial feedback.

Indeed, the individual degree of history bias adjustment made a significant contribution to individual performance (*Figure 1F*). We computed an individual measure of bias adjustment from the weights of both previous stimulus and choice (Materials and methods) and used this to predict participants' overall task performance in the structured environments (proportion of correct choices collapsed across repetitive and alternating). As expected, individual performance also strongly depended on participants' sensitivity to the current evidence (i.e., slope of the psychometric function). We, therefore, used partial regression to quantify the unique contribution of each factor (history bias adjustment and evidence sensitivity) on performance. Both factors uniquely predicted performance (sensitivity: $r$=0.798, p<0.0001; bias adjustment: $r$=0.493, p=0.0026; Pearson correlation), with a clear effect of the adjustment of history biases (*Figure 1F*). The same was true when we used the individual weights of the previous stimulus for performance prediction, separately for the two biased environments, but not the neutral environment (*Figure 1—figure supplement 3*).

## Large-scale cortical dynamics of task processing

The behavioral results reported above indicate that participants adjusted their history bias to the environmental statistics, which, in turn, boosted their performance. How did these (partly) adaptive history biases affect the formation of subsequent decisions, more specifically: the dynamics of the underlying DV in the brain? Our concurrent collection of whole-brain MEG data during this task enabled us to address this question. We combined source reconstruction with established anatomical atlases and spectral analysis to characterize the cortical dynamics involved in our task across several pre-defined cortical regions known to be involved in visual processing and action planning (*Wilming et al., 2020*; *Murphy et al., 2021*).

We first identified established task-related modulations of MEG power during decision formation. Control analyses indicated minor leakage between the source estimates for neighboring cortical regions, but negligible leakage for more distant regions (*Figure 2—figure supplement 1*; Materials and methods). Importantly, our analysis revealed distinct functional profiles for several regions, in line with previous work (*Siegel et al., 2011*; *Wilming et al., 2020*; *Murphy et al., 2021*; *Urai and Donner, 2022*): modulations in visual cortical regions that scaled with motion coherence and encoding of the evolving action plan in frontal (motor and premotor) and parietal cortical regions (*Figure 2*).

In line with previous work (*Siegel et al., 2007*), gamma-band power (~60–100 Hz) in visual cortex was enhanced while low-frequency power (<30 Hz) was suppressed relative to baseline during motion viewing (*Figure 2A*); both components of the visual responses scaled with motion coherence, predominantly in dorsal visual cortical areas V3A/B, IPS0-3, and the MT+ complex (*Figure 2B*). Concomitantly with these responses to visual motion, activity lateralization predicting the subsequent choice (left vs. right button) built up in downstream (anatomically more anterior) parietal and motor cortical areas (*Figure 2C*). Again in line with previous work (*Donner et al., 2009*; *de Lange et al., 2013*; *Pape and Siegel, 2016*; *Wilming et al., 2020*; *Murphy et al., 2021*), this action-selective activity build-up was a suppression of beta-band (12–36 Hz) power contra- vs. ipsilateral to the upcoming movement, and robustly expressed in the M1 hand area (*Figure 2C* and Figure 4A). This signal, referred to as 'motor beta lateralization' in the following, has been shown to exhibit hallmark signatures of the DV, specifically: (i) selectivity for choice and (ii) ramping slope that depends on evidence strength (*Siegel et al., 2011*; *Murphy et al., 2021*; *O'Connell and Kelly, 2021*).

This signal reached statistical significance earlier for correct than error trials and during the stimulus interval it ramped to a larger amplitude (i.e., more negative) for correct trials (*Figure 2—figure supplement 2*, left). But the signal was indistinguishable in amplitude between correct and error trials around the time of the motor response (*Figure 2—figure supplement 2*, right). We also confirmed the dependence of the ramping of the motor beta lateralization on evidence strength using a single-trial regression also including the history bias that we report in the section Adaptive history bias shapes the build-up of action-selective motor cortical activity below.

In sum, we replicated well-established signatures of visual motion processing and action-selective motor preparation in our current MEG data – for the latter signal in particular, some hallmark signatures of the DV: selectivity for choice, dependence of slope on evidence strength, and dependence

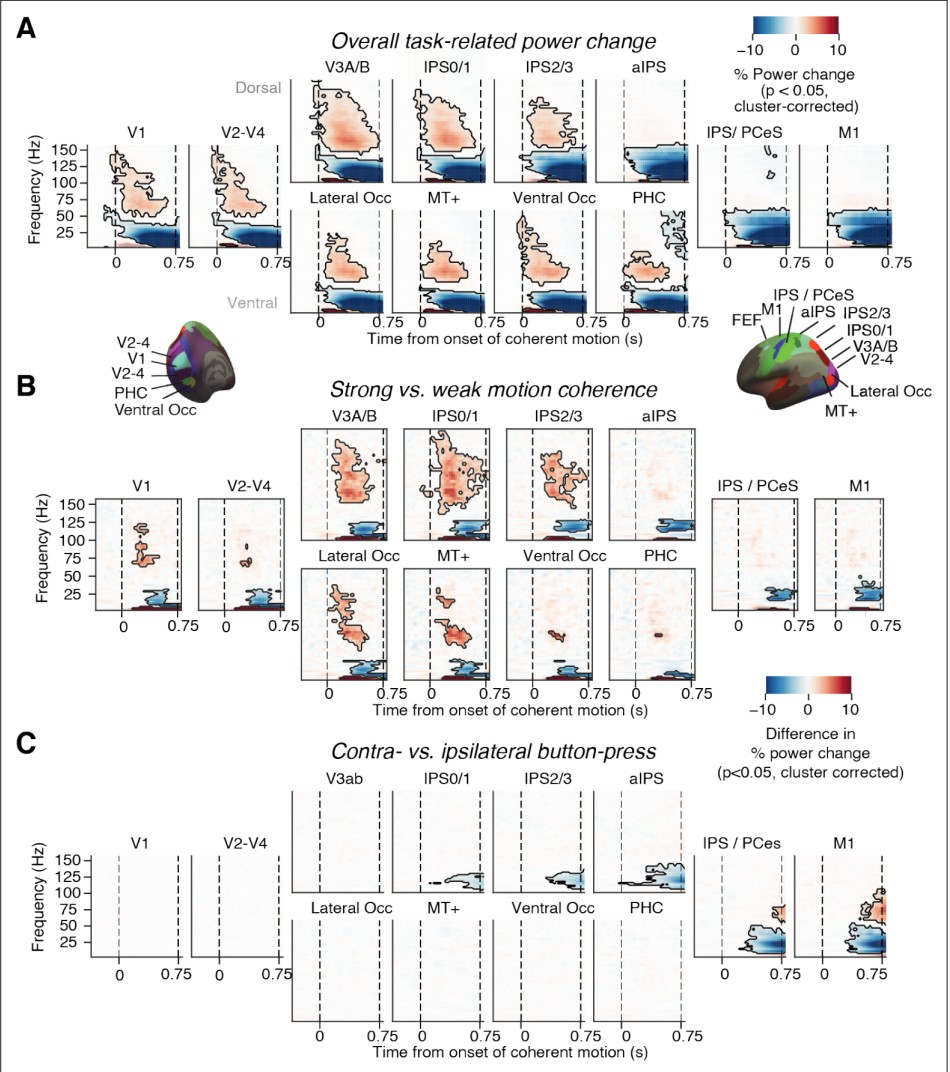

**Figure 2.** Neural signatures of stimulus processing and action planning across the cortical visuo-motor pathway. (**A**) Overall task-related power change (average across hemispheres). Increase in visual gamma-band response and decrease in alpha- and low-beta-band power in visual cortex during presentation of coherently moving dots. (**B**) Motion coherence-specific sensory response. Difference in time-frequency response between high (0.81%) and 0% motion coherence (average across hemispheres). Increase in visual gamma-band power and decrease in alpha- and low-beta-band power scale with motion coherence of stimulus. (**C**) Time-frequency representation of action-selective power lateralization contralateral vs. ipsilateral to upcoming button-press. All signals are expressed as percentage of power change relative to the pre-trial baseline. Dashed vertical lines, onset and offset of coherent motion. Saturation, significant time-frequency clusters (p<0.05, two-tailed cluster-based permutation test).

The online version of this article includes the following figure supplement(s) for figure 2:

**Figure supplement 1.** Correlation of linearly constrained minimum variance (LCMV) beamformer weights.

**Figure supplement 2.** Action-selective motor cortical activity ramps up to a larger amplitude during the stimulus interval for correct vs. error but converges at same level before choice.

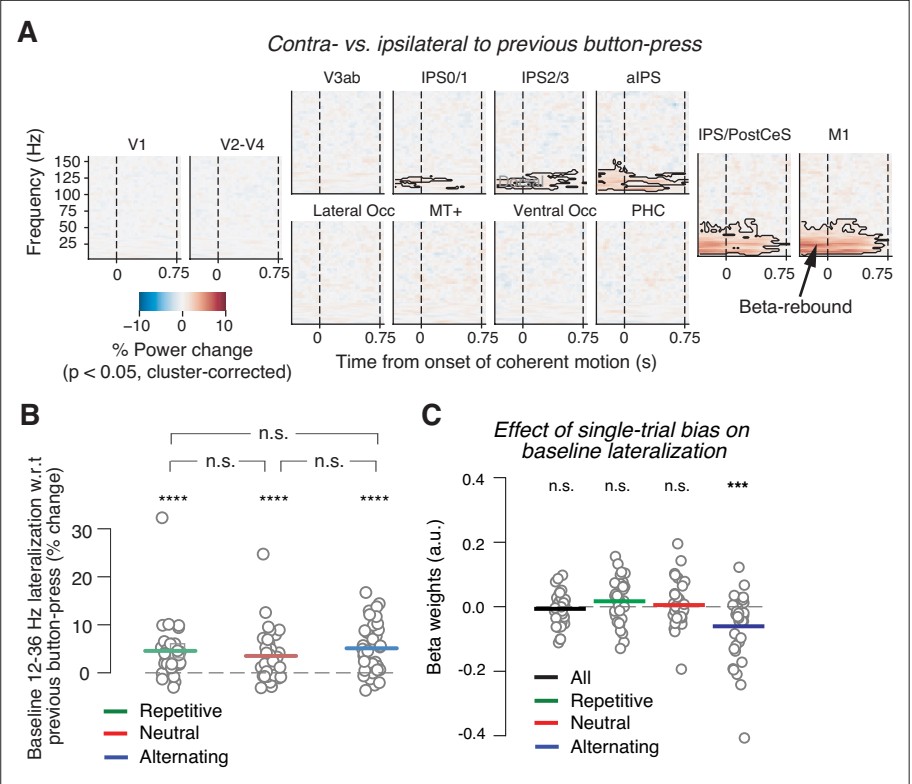

**Figure 3.** Baseline state of motor cortex reflects previous choice, but not consistently context or history bias. (**A**) Spill-over of action-selective beta-power rebound from previous into current trial. Time-frequency representation of power lateralization contra- vs. ipsilateral to the previous button-press, expressed as percentage power change from baseline. Enhanced beta-band power contra- vs. ipsilateral to the previous button-press in motor cortices. Dashed vertical lines mark the onset and offset of coherent motion. Saturation, significant time-frequency clusters (p<0.05), two-tailed cluster-based permutation test across participants. (**B**) Impact of sensory environment on overall baseline state of beta lateralization (350 to 100 ms before stimulus onset) contra- vs. ipsilateral to previous button-press. (**C**) Impact of single-trial history bias on amplitude of M1 beta lateralization (relative to up-coding hand) during baseline interval (from 350 to 100 ms before evidence onset). ***p<0.001, ****p<0.0001 (two-tailed permutation test).

of behavioral accuracy. In what follows, we systematically quantify the impact of the adaptive history bias on two fundamental aspects of this neural DV marker: its baseline level at the start of the decision process and its build-up during decision formation.

## No consistent modulation of baseline state of action-selective activity by environmental context and trial history

Previous human MEG work indicates that the motor cortical baseline beta-power state is flipped relative to its state just before the previous choice, a phenomenon referred to as 'beta rebound' (*Pfurtscheller et al., 1996*; *Pape and Siegel, 2016*; *Urai and Donner, 2022*) that was also evident in our data (*Figure 3A*, collapsed across all three environments). Recent MEG studies of human perceptual decision-making have linked this phenomenon to either overt choice alternation (*Pape and Siegel, 2016*) or alternating starting points inferred from drift diffusion model fits (*Urai and Donner, 2022*). We, therefore, wondered if and how the baseline level of motor beta lateralization depended on the different sensory environments or on the history bias in specific trials.

If this baseline lateralization state 'inherited from' the previous choice was involved in mediating the effect of the adaptive bias on choice, it might be expected to be reduced in repetitive vs. alternating environments, thus reducing subjects' tendency to alternate in the former. Instead, the beta rebound effect (i.e., increased power contralateral vs. ipsilateral to previous choice) was about equally strong in all three environments (*Figure 3B*). We found no evidence for its modulation by sensory

environment (repetitive vs. neutral: p=0.1511; repetitive vs. alternating: p=0.5667; neutral vs. alternating: p=0.0789; all two-sided permutation tests).

We then related the baseline M1 beta lateralization to the adaptive history bias. To this end, we adapted a single-trial regression procedure from recent monkey physiology work (*Mochol et al., 2021*) to relate the time-varying history bias to neural data in order to test if this bias modulated the baseline motor beta lateralization on a trial-by-trial basis. We used each individual's time course of single-trial history bias estimated through the behavioral model (positive values for bias toward upward, *Figure 1E*) as predictors for the single-trial motor beta lateralization, whereby lateralization was assessed relative to the hand coding for up-choices in a given block (Materials and methods). This procedure took the impact of both previous stimuli and previous choices into account, estimated with an individually optimized number of lags. Thus, the single-trial bias estimates were largely independent of assumptions about the sources of the single-trial bias (stimuli, choices, lags). However, because the model was fit and applied separately to data from different environments, the resulting time course of single-trial bias estimates captured the context-dependent, adaptive bias components described in the preceding section. An involvement of the motor baseline state in the implementation of history bias predicts a stronger baseline beta-suppression contralateral to the hand favored by the bias, with a magnitude that scales with the strength of the bias. In other words, this scenario predicts significant negative beta coefficients, regardless of the environment.

We found no such effect when the analysis was run across all three environments (*Figure 3C*, 'all'), again inconsistent with the notion of a generally bias-encoding neural signal. We did find an effect of the single-trial bias on the baseline beta lateralization state in the alternating environment when analyzed selectively (*Figure 3C*; p=0.0003, two-tailed permutation test). Such an effect was, however, not present for either of the other two environments (*Figure 3C*; all: p=0.4465; repetitive: p=0.1343; neutral: p=0.6571; two-tailed permutation tests). Overall, the results of our analyses of the baseline beta lateralization suggest the beta rebound from the previous trial may help promote choice alternation when performing in an alternating context, but does not generally encode adaptive stimulus history biases.

## Adaptive history bias shapes the build-up of action-selective motor cortical activity

The analyses from the previous section assessed the dependence of the starting point (i.e., baseline level) of a neural DV-proxy on environmental context and adaptive history bias. Behavioral modeling has shown that idiosyncratic history biases in a variety of tasks in random environments are accounted for by history-dependent biases in the build-up (i.e., drift) of the DV, rather than in its starting point (*Urai et al., 2019*). We, therefore, next asked whether the adaptive history biases identified here might shape the build-up rate of our neural proxy of the DV during decision formation.

To test this idea, we used two complementary approaches both of which again exploited our model-inferred single-trial bias estimates. In one of those approaches, we grouped the single-trial bias estimates into three equally spaced bins, with two bins containing strong biases of opposite direction (up vs. down) and the middle bin containing trials with little bias (*Figure 1E*). We used the 'up' and 'down' bins to visualize the impact of the history bias on the ramping of the neural DV, by computing the time course of beta lateralization contra- vs. ipsilateral to the button-press for the direction of the bin-wise bias (Materials and methods). The behavioral choice was, by definition, correlated with both, the single-trial bias and the action-selective motor beta lateralization (*Figure 2C* and *Figure 4A*). This could yield a correlation between bias and motor cortical lateralization even in the absence of any direct effect of bias on motor beta lateralization. To isolate a genuine effect of the bias on our neural DV, we subsampled the data from the up and down bias bins to yield an equal number of upward and downward choices within each bin (Materials and methods). For each bin, we then computed the time course of beta lateralization relative to the button-press for the bias and collapsed the resulting time courses across bins. This procedure isolated the impact of the model-inferred history bias on the neural DV, independent of the choice. The resulting time course ramped into the direction of the single-trial bias, reaching statistical significance at about 700 ms after motion onset, before the end of the decision interval (*Figure 4B*). We used linear regression to estimate the slope of the ramp for an interval that exhibited clear linear ramping in the average motor beta lateralization across all trials (gray-shaded in *Figure 4A*, Materials and methods). As expected, the slope was smaller than zero

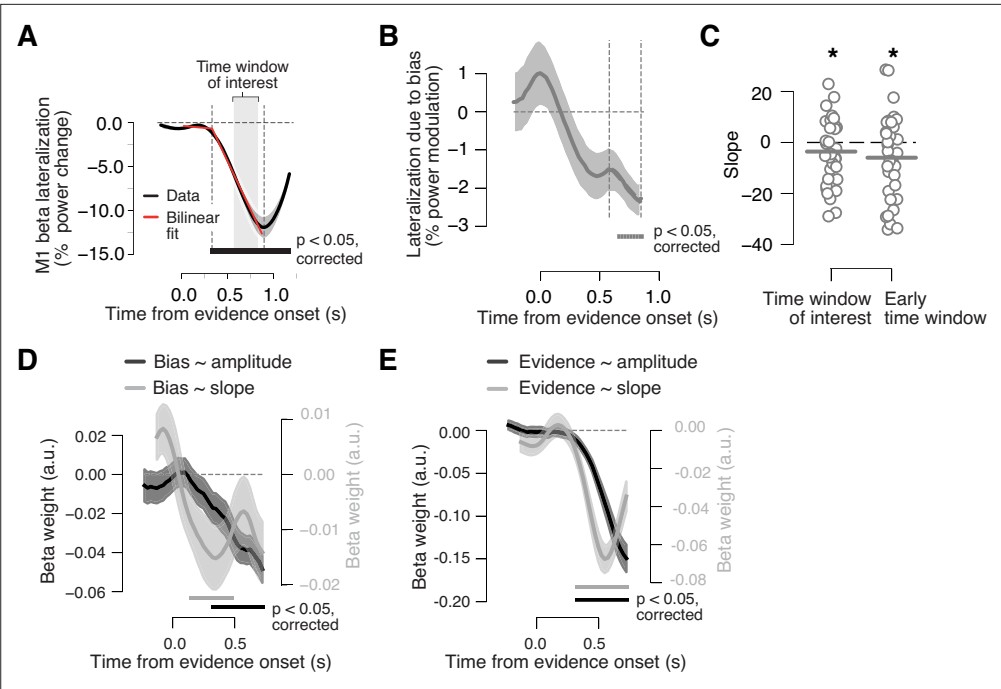

**Figure 4.** Adaptive biasing of action-selective build-up activity in M1. (**A**) Time course of action-selective beta-power (12–36 Hz) lateralization in the M1 hand area, contralateral vs. ipsilateral to upcoming button-press, collapsed across trials (black line). Red line, bilinear fit. Gray box, time window (0.58 to 0.8475 s from evidence onset) used to quantify the (rate of) build-up of power lateralization in panels **B and C** (vertical dashed lines in **B**). The window was defined to start 250 ms after the intersection point of bilinear fit and end 50 ms before the minimum of power lateralization, chosen so as to cover the interval containing ramping activity in the majority of trials. (**B**) Component of action-selective lateralization governed by single-trial bias, irrespective of upcoming behavioral choice and pooled across sensory environments (see main text for details). (**C**) Slope estimates for neural bias measures from panel B. Left, time window from panel A. Right, early time window derived from single-trial regression in panel **D**. (**D**) Time-variant impact of single-trial history bias on amplitude (black) and slope (gray) of M1 beta lateralization (relative to up-coding hand). (**E**) Same as D but for impact of signed stimulus strength. N = 36; shaded areas, SEM. Bars, p<0.05 (two-tailed cluster-based permutation test) across participants. *p<0.05 (one-tailed paired permutation test).

The online version of this article includes the following figure supplement(s) for figure 4:

**Figure supplement 1.** Subsampling procedure does not change distribution of coherences.

(p=0.048; one-tailed permutation test against zero; *Figure 4C*, left), indicating that the time-varying history bias contributed to the build-up of action-selective motor cortical activity during decision formation. We obtained qualitatively identical results as in *Figure 4B and C* when first removing (using linear regression) the beta-rebound from the previous trial (*Figure 3A*) from the time course of the beta-band lateralization (data not shown).

Second, we again fit a single-trial regression model, now to simultaneously quantify the impact of the history bias and current evidence on the dynamics of the neural DV in a time-variant fashion. We ran two separate regression models, one on the amplitude of motor beta lateralization for a range of time windows, the other on the slope of motor beta lateralization, assessed locally in time for the same time windows; in both cases, lateralization was again assessed relative to the hand coding for up-choices in a block (Materials and methods). An impact of the adaptive history bias on the ramping of motor cortical activity would predict a specific effect of the history bias, over and above the effect of the sensory evidence, on both read-out measures, in particular on the ramping slopes. Specifically, it predicts negative beta weights, reflecting steeper downward slope (i.e., stronger suppression) for stronger biases.

We found a clear and expected effect of current sensory evidence on motor beta lateralization, with a steeper downward slope for stronger evidence (*Figure 4E*). This dependence of the ramping of

the motor beta lateralization on current motion strength is in line with previous work (*de Lange et al., 2013*) and confirms that the motor beta lateralization reflects hallmarks of a neural DV. Critically, and in line with our hypothesis, the same was true for the effect of the history bias: a stronger bias produced a stronger and steeper motor beta lateralization toward the direction of the bias (*Figure 4D*). The bias effect on the lateralization amplitude reached significance during the decision interval (from about 320 to about 720 ms after motion onset; *Figure 4D*, black line), and the corresponding impact on the ramping slope was significant even earlier, during the first half of the decision interval (starting at about 150 ms after the motion onset; *Figure 4D*, gray line). Combined, these two effects indicate that a strong bias on a given trial constituted an early force on the M1 ramping dynamics, pushing the signal into the direction of the bias even before the current evidence exerted its effect (compare with gray lines in *Figure 4E*); the M1 lateralization amplitude later during the decision interval reflected the bias more strongly on trials, for which the bias was strong than those, for which the bias was weak.

Our analysis of the ramping slopes in *Figure 4C* (left) estimated the slope for a longer (and later) time interval than the one, for which the single-trial regression in *Figure 4D* yielded significant slope effects. We, thus, repeated the above analysis also for the earlier time window derived from the single-trial regression results *Figure 4C* (right). Also, for this window did we find a robust effect of the bias estimate on the ramping slope (p=0.0158; one-tailed permutation test against zero). Taken together, model-independent and model-based analyses provided convergent evidence for the dependence of action-selective cortical ramping activity during decision formation on the time-varying, context-dependent history biases.

## Discussion

It has long been known that the history of preceding choices and stimuli biases perceptual judgments of the current stimulus (*Fernberger, 1920*). Recent behavioral modeling showed that at least part of such history biases reflect time-varying expectations that are flexibly adapted to the environmental structure (*Abrahamyan et al., 2016*; *Braun et al., 2018*; *Hermoso-Mendizabal et al., 2020*). Such dynamically varying expectations, largely ignored in standard neurophysiological studies of perceptual decision-making, may be a key driver of sensory-guided behavior in ecological settings (*Mobbs et al., 2018*). How adaptive expectations shape the neural dynamics underlying decision-making has remained unknown. Here, we addressed this issue by combining a standard task from the neurophysiology of decision-making (*Gold and Shadlen, 2007*; *Siegel et al., 2011*) with systematic manipulations of the environmental stability as well as single-trial, model-based MEG assessment of cortical decision dynamics. This revealed that the history-dependent, dynamic expectations boosted participants' behavioral performance and selectively altered the build-up sign and rate, not (consistently) the pre-trial baseline level, of an established neurophysiological proxy of the DV: action-selective preparatory population activity in their motor cortex.

While participants' history biases in a random environment (i.e., uncorrelated stimulus sequences) were largely idiosyncratic, as widely observed (*Akaishi et al., 2014*; *Urai et al., 2019*; *Urai and Donner, 2022*), we found that one component of these biases lawfully shifted between stable (frequent category repetitions) and systematically alternating environments and improved participants' performance. It is instructive to compare this adjustment of history bias with the one observed in a previous study using a similar manipulation of environmental statistics (*Braun et al., 2018*). In that previous study, participants did not receive outcome feedback and thus remained uncertain about the category of the previous stimulus. Correspondingly, the history bias adjustment was evident in the impact of their previous choices (rather than previous stimulus categories), and most strongly of those made with high confidence (i.e., correct and fast). By contrast, in the current study, participants could deterministically infer the true category of the previous stimulus from the feedback. Correspondingly, we found that their history bias adjustment to the different environments was now governed by the previous stimulus category. Together, the findings from both studies support the notion that human subjects can use different types of internal signals to build up history-dependent expectations in an adaptive fashion.

The observation of an effective behavioral adjustment to differentially structured environments in participants' steady-state behavior raises the question of how (and how quickly) they learned the different environmental structures. Our behavioral modeling approach required many trials, which precludes the assessment of the temporal evolution of the bias (i.e., weight) adjustment during the

blocks of a given sensory environment. This issue should be addressed in future work, using models capable of learning environmental parameters such as transition probabilities (*Yu and Cohen, 2009*; *Meyniel et al., 2016*; *Glaze et al., 2018*; *Hermoso-Mendizabal et al., 2020*).

Previous work has characterized neural signals underlying idiosyncratic history biases in contexts where these biases would be maladaptive. Such signals were observed in several brain areas and in different formats. In a continuous spatial working memory task, activity-silent codes in prefrontal cortex during mnemonic periods seem to promote memory reactivations, which mediate serial memory biases (*Barbosa et al., 2020*). Studies of perceptual forced choice tasks have found signatures of persistent population activity reflecting the previous choice in posterior parietal cortex (*Morcos and Harvey, 2016*; *Hwang et al., 2017*; *Scott et al., 2017*; *Urai and Donner, 2022*), prefrontal cortex (*Mochol et al., 2021*), and motor cortex (*Pape and Siegel, 2016*; *Urai and Donner, 2022*). Specifically, human MEG work showed that history-dependent modulations of parietal cortical activity in the gamma-band spanned the intervals between trials and mediated idiosyncratic choice repetition biases (*Urai and Donner, 2022*). Such an effect was not observed for the motor beta-rebound that was similarly sustained into the next decision interval (*Urai and Donner, 2022*). Importantly, none of these studies quantified the build-up rate of action-selective motor cortical activity on the subsequent trial.

Idiosyncratic history biases are reflected in a persistent baseline state of action-selective neural population activity in monkey prefrontal cortex, during decision formation accompanied by a subtler modulation of the build-up rate (*Mochol et al., 2021*). Another human MEG study derived an action-independent proxy of the neural DV from sensor-level MEG data that required two successive judgments within a trial (*Rollwage et al., 2020*). The initial decision biased the subsequent build-up of that DV-proxy in a manner that depended on the consistency of new evidence with the initial decision and the confidence in that initial decision. Critically, no previous study has investigated the flexible and performance-increasing history biases that we have manipulated and studied here.

Our current results resemble the results from block-wise manipulations of the probability of a specific stimulus category (i.e., not of transitions across stimulus categories): this also biases the build-up of saccade-selective activity in monkey posterior parietal cortex (*Hanks et al., 2011*), just like what we found here for hand movement-selective motor cortical activity in humans, albeit with strong, but lawful, trial-by-trial variations in our current setting (*Figure 1E*). It is tempting to interpret both as downstream expressions of perceptual expectations in cortical circuitry involved in action planning. Indeed, modulating the build-up of an evolving DV by prior expectations can be useful in accumulation-to-bound models when reliability of the evidence varies from decision to decision (*Hanks et al., 2011*; *Moran, 2015*). Whether or not the neural signatures of idiosyncratic history biases studied in previous work have similar cognitive content and underlying mechanisms remains an open question.

Our results indicate that dynamic and adaptive expectations bias the dynamics of neural signatures of action planning during decision formation. How are these expectations implemented in upstream neural populations, so as to yield the selective changes in M1 ramping dynamics observed here? One possibility is that history biases the state of sensory cortex (*Nienborg and Cumming, 2009*; *St John-Saaltink et al., 2016*), for example via feedback from cortical areas involved in decision formation (*Wimmer et al., 2015*). Another possibility is that the expectations shape the read-out of sensory evidence by the evidence accumulator, with preferential accumulation of evidence that matches the expectation, in line with active inference (*Friston, 2010*). Yet another possibility is that the evidence accumulator receives non-sensory input from brain regions encoding history information in a sustained fashion (*Talluri et al., 2021*; *Urai and Donner, 2022*). In all these different schemes, dynamic expectations would need to be constructed in a highly flexible, context-dependent fashion in order to give rise to the adaptive biasing of action-selective activity observed here.

## Materials and methods
### Participants
42 healthy human observers (27 female, 15 male, $18 \leq$ age $< 40$) participated in the study. The sample size was based on a previous psychophysical study with similar behavioral task (*Braun et al., 2018*). Inclusion criteria were no history of neurological and psychiatric illness and an age between 18 and 40 years. All participants did not meet any of the standard exclusion criteria for MEG and

MRI recordings (pregnancy, claustrophobia, pacemaker or other implanted biomedical devices, non-MRI- or MEG-compatible metallic implants or foreign bodies in the body, hearing disorder, impaired temperature sensation and/or increased sensitivity to heating of the body, rejection of information about unexpected morphologic findings in anatomic MRI measurement) and gave their written informed consent. The experiment was approved by the local ethical review board (Ärztekammer Hamburg reference number PV4714). Two participants showed performance around chance level in the training session and therefore did not participate in the MEG sessions. Two more participants were excluded from the analysis so that 38 participants remained for the data analysis. One of the excluded participants did not respond within the response interval on a substantial number of trials (31% of trials), and the other participant was excluded due to excessive MEG artifacts. We excluded three recording sessions (from different participants) due to substantially worse-than-average behavioral performance or missing data files. One participant completed only one MEG session.

## Behavioral task

We used a random dot motion discrimination task with varying levels of evidence strength (motion coherence) spanning the psychophysical threshold (*Figure 1A*). Participants had to judge whether a cloud of coherently moving signal dots embedded in dynamic noise was either moving upward or downward. To interrogate the adaptability of trial history biases, participants performed the task in three different stimulus environments, defined by varying levels of autocorrelation between stimulus categories (upward or downward) across trials. In a 'neutral' environment the direction of motion was chosen at random on each trial, in a 'repetitive' environment the previous motion direction was more likely to be repeated (80% repetition probability) and in an 'alternating' environment the previous motion direction was more likely to be alternated (20% repetition probability). The resulting fractions of upward and downward motion stimuli were approximately equal within each environment: The group average frequency of upward trials was 0.502 for neutral, 0.507 for repetitive, and 0.500 for alternating.

The MEG data of this experiment allowed for identifying the neural correlates underlying the history bias adjustment.

## Stimuli

Random dot kinematograms contained 117 white dots at a density of 6 dots/deg$^2$ on a gray screen. Each dot had a size of 0.06°. The dots were moving within a circular aperture of 2.5° radius of visual angle centered around a fixation cross of 0.2° × 0.2°. The aperture was placed 3.5° below the center of the screen. Random dots (0% coherence) were presented throughout the whole trial to guarantee constant luminance in order to avoid luminance-induced changes in pupil diameter. During the evidence interval, coherently moving signal dots were superimposed onto the random noise dots. The signal dots moved either upward or downward (or in random directions in case of 0% motion coherence). The motion coherence, that is, the percentage of coherently moving dots, was chosen from trial to trial at random out of five levels (0%, 3%, 9%, 27%, 81%) under the constraint that each block contained an equal number of trials per motion coherence and direction. The signal dots were moving with a velocity of 11.5°/s and each dot had a lifetime of 10 frames. Three variants of dot motion (at the same coherence and direction) were presented in an interleaved fashion within each trial.

## Trial structure

The fixation cross changed its color to indicate different periods within each trial. Each trial started with a fixation interval of 0.75–1.5 s (uniformly distributed), during which the fixation cross was colored in red. After the fixation interval, the fixation cross turned green to indicate the onset of coherent motion. After a fixed evidence duration of 0.75 s, the signal dots disappeared from the screen and the fixation cross turned red again to indicate the start of the response interval. Participants were instructed to report their choice with a left- or right-hand button-press. The choice-hand mapping was counterbalanced within each participant and randomly chosen per block with the restriction that both choice-hand mappings occurred once per stimulus environment per session. After button-press or a maximum response time of 1.25 s in case no response was given, the fixation cross turned blue and the inter-trial interval started. After a uniformly distributed interval of 1.5–2.5 s (pupil rebound time after response), participants received auditory feedback (0.15 s) about the accuracy of their response.

A high tone (1100 Hz) was given for a correct response, a low tone (150 Hz) for an incorrect response, an intermediate tone (440 Hz) after a 0% coherence trial (accuracy not defined) and a white noise tone if the participant did not respond within the maximum response time. The inter-trial interval continued for another 2–2.5 s (uniformly distributed). Participants were instructed to fixate the cross during the entire trial and not to blink during all periods but the inter-trial interval.

Participants performed one training session and three MEG sessions of 2 hr each. Each session consisted of 6 blocks of 99 trials each. The repetition probability between the two motion directions remained constant within each block but randomly varied across blocks under the constraint that each session contained two blocks of each environmental condition. Participants were not informed about the manipulation of the stimulus sequence.

## Behavioral modeling of trial history bias
### Logistic regression model with history bias

To quantify the influence of the history of previous choices and stimulus categories on the current choice, we used a logistic regression model with a history-dependent bias term that shifted the psychometric function along the horizontal axis (*Fründ et al., 2014*; *Urai et al., 2017*; *Braun et al., 2018*). Specifically, the probability of making one of the two choices $c_t = 1$ ($c_t = 1$ for 'choice up', $c_t = -1$ for 'choice down') on trial $t$ was described by:

$$P\left(r_t = 1 | \widetilde{s_t}, \boldsymbol{h_t}\right) = \gamma + \left(1 - \gamma - \lambda\right) g\left(\delta\left(\boldsymbol{h_t}\right) + \alpha \widetilde{s_t}\right). \tag{1}$$

and $\lambda$ were the lapse rates for the choices $c_t = 1$ and $c_t = -1$, and $g\left(x\right) = \frac{1}{1+e^{-x}}$ was the logistic function. $\widetilde{s_t}$ was the signed stimulus intensity (i.e., motion coherence times stimulus category; 'up' or 'down', coded as 1 and −1) and $\alpha$ was the slope of the stimulus-dependent part of the psychometric function, quantifying perceptual sensitivity. The bias term

$$\delta\left(\boldsymbol{h_t}\right) = \delta' + \delta_{hist}\left(\boldsymbol{h_t}\right) = \delta' + \sum_{k=1}^{2n} \omega_k h_{kt} \tag{2}$$

that is, the offset of the psychometric function, consisted of an overall bias $\delta'$ for one specific choice ('up' or 'down') and a history-dependent bias term $\delta_{hist}\left(\boldsymbol{h_t}\right) = \sum_{k=1}^{2n} \omega_k h_{kt}$, which was the sum of the preceding $n$ (see Determination of model order below for determination of $n$) choices $c_{t-1}$ to $c_{t-n}$ and the preceding $n$ stimulus categories $z_{t-1}$ to $z_{t-n}$, each multiplied with a weighting factor $\omega_k$. The vector $\boldsymbol{h_t}$ was made up of the last $n$ choices and stimulus categories: $\boldsymbol{h_t} = \left(c_{t-1}, \ldots, c_{t-n}, z_{t-1}, \ldots, z_{t-n}\right)$. Upward and downward choices and stimulus categories were coded as 1 and −1 and stimuli with zero motion coherence were set to 0. The weighting factors $\omega_k$ specified the influence of each of the $n$ preceding choices and stimulus categories on the current choice. Positive values of $\omega_k$ referred to a tendency to repeat, and negative values of $\omega_k$ referred to a tendency to alternate the choice or stimulus category at the corresponding lag. All parameters were fit by maximizing the log-likelihood $L = \sum_t log\left(P\left(r_t = 1 | \widetilde{s_t}, \boldsymbol{h_t}\right)\right)$ using an expectation maximization algorithm (*Fründ et al., 2014*). The slope was fitted separately for each session and then averaged across sessions.

In *Figure 1—figure supplement 2A*, we tested the clustering of vector angles of the shift between the weights from neutral and the weights from the repetitive or alternating environments, respectively, and the difference of these shifts between both environments. The same qualitative pattern of results was observed when the shift angles for repetitive and alternating environments were computed with respect to the origin rather than the individual data points for neutral.

In *Figure 1F*, we computed an individual measure of bias adjustment as the length of the vector between the weights from repetitive and alternating from *Figure 1—figure supplement 2A*.

## Determination of model order

To avoid overfitting, we determined the model order, that is, the number of lags $n$ in the logistic regression model that described the behavioral data best, separately for each subject and each environmental condition using a sixfold cross-validation procedure. We split the data into six test and training sets. Each test set contained one out of the six blocks of each environment, and the training set contained the remaining five blocks. We shuffled the assignment of the test block and

the training blocks across all six possibilities resulting in six different pairs of test and training datasets. For each training dataset, we fitted the logistic regression model with varying number of lags ranging from 0 (no history) to 7 lags. For each fold and model order, we computed the log-likelihood $L = \sum_t log \left( P \left( r_t = 1 | \widetilde{s_t}, \boldsymbol{h_t} \right) \right)$ using the choices and stimuli from the test data and the fitted model parameters, that is, history weights, general bias, lapse rate, and slope from the corresponding training data. We averaged the log-likelihood values for each subject and model order across the six folds. The model with the maximum log-likelihood value defined the best fitting model order $n$ that was used for the subsequent analyses (*Figure 1—figure supplement 1*). For those subjects for which the model without history bias, that is, zero lags, was the best fitting model for one biased environment, we set the model order for the corresponding environment to 1 for the behavioral analyses. We excluded two subjects from the analyses of the MEG data for which the model without history bias, that is, zero lags, was the best fitting model for both biased environments, as those subjects did not adapt their choice behavior to the statistical structure of the environment.

## Single-trial bias estimates

To obtain an estimate of the bias at each single trial (*Figure 1E*), we computed the bias term $\delta \left( h_t \right) = \delta' + \delta_{hist} \left( h_t \right) = \delta' + \sum_{k=1}^{2n} \omega_k h_{kt}$ using the vector of previous choices and stimulus categories $\boldsymbol{h_t}$ from each test dataset (block) and the general bias $\delta'$ and history weights $\omega_k$ for the previous choices and stimulus categories at lag $k = 1$ to $n$ fitted from the corresponding training dataset. By fitting the model excluding the block from the test dataset, we guaranteed that the single-trial bias estimates were not contaminated by the data that they were supposed to predict. The sign of the single-trial bias $\delta$ determined the tendency for an 'up' (for a positive sign) or 'down' (for a negative sign) choice before stimulus presentation (different from the history weights $\omega_k$, which indicate a tendency to repeat or alternate). The magnitude of the single-trial bias $\delta$ defined the strength of this tendency.

We binned the single-trial bias estimates into three bins of equal size separately for each subject. The low bin contained the values in the 0–33% quantile, the medium bin contained the values in the 33–66% quantile, and the high bin contained the values in the 66–100% quantile. On average, the values in the low bin were negative corresponding to a bias for choice 'down', the medium bin contained a bias close to zero, and the values in the high bin were positive indicating a bias for choice 'up'.

## MEG data acquisition and analysis

### Data acquisition

MEG data was recorded with a whole-head 275-channel CTF system at a sampling rate of 1200 Hz. We simultaneously recorded saccades and pupil dilation using an EyeLink 1000 Long Range Mount (SR Research, Osgoode, Ontario, Canada) and vertical and horizontal EOG as well as a bipolar electrocardiogram using Ag/AgCl electrodes. To monitor the subjects' head position, we used three fiducial coils: one above the nasion and one each in the left and right auricle. We used online head-localization (*Stolk et al., 2013*) to adjust the subjects' head position before each block to maintain the same head position relative to the MEG sensors across blocks within each session. To obtain the same head position across all three MEG sessions, we located the subjects' head position in the second and third session relative to its position in the first session. Stimuli were shown on a screen with a refresh rate of 60 Hz, at a distance of 65 cm from the subjects' eyes using a beamer with a resolution of 1024×768 pixels.

## Preprocessing

First, the data was down-sampled to 400 Hz and epoched into single trials from fixation (0.75 s before the evidence interval) to 1.5 s after feedback. Then, we cleaned the data from artifacts via visual inspection as well as through semi-automatic artifact rejection routines using the Fieldtrip Toolbox (*Oostenveld et al., 2011*). We removed trials in which no response was given within the maximum response interval of 1.25 s after evidence offset and trials with excessive head motion >6 mm deviation from the first trial (*Stolk et al., 2013*). We removed line noise around 50, 100, and 150 Hz using a bandstop filter and demeaned and detrended the data. To detect artifacts caused by cars passing by the MEG lab, we low pass filtered the data at 1 Hz, applied a Hilbert transform, z-scored

the data, and removed trials with large amplitudes and a slow drift of the resulting signal via visual inspection. Muscle bursts and squid jumps were detected via visual inspection after applying a ninth order 110–140 Hz Butterworth filter, a Hilbert transform, and z-scoring. Eye blinks and saccades were identified via visual inspection of the vertical and horizontal EOG channels after applying a 1–15 Hz bandpass filter, a Hilbert transform, and z-scoring the data. Trials with muscle bursts, eye blinks, or saccades were removed in case those artifacts occurred before the response. The cleaned data was epoched into stimulus-locked (−0.55 to 1.5 s around evidence onset) and response-locked (−0.5 to 1.5 s around button-press) segments.

## Spectral analysis

Single-trial complex time-frequency representations of the source-reconstructed signal were computed with a window length of 400 ms in steps of 25 ms using MNE (*Gramfort et al., 2014*). For the low frequencies (3–37 Hz in steps of 2 Hz), we used one taper and a frequency smoothing of 5 Hz (2.5 Hz half window). For the high frequencies (37–161 Hz in steps of 4 Hz), we used a multitaper approach (using Morlet wavelets windowed with discrete prolate spheroidal sequences) with seven tapers and a frequency smoothing of 20 Hz (10 Hz half window). Then, the beamformer weights (see next section) of the vertices within each region of interest (ROI) were applied to the complex output of the time-frequency representations before computing the power and averaging across trials and vertices. For each ROI and frequency, we computed the baseline as the average power across trials during the interval ranging from 350 to 100 ms before evidence onset, separately for each subject and session. The data for each ROI and frequency was then transformed into percent signal change from the corresponding baseline.

## Source reconstruction

We used linearly constrained minimum variance (LCMV) beamforming (*Van Veen et al., 1997*) and time-frequency decomposition to reconstruct the local field potentials at the source level. We first reconstructed the cortical surface from each participant's anatomical MRI scan using freesurfer (*Dale, 1999*; *Fischl et al., 1999*). In case no MRI scan was available (3 subjects), we used an average subject provided by freesurfer, that was obtained from the average across 40 subjects. Then, we aligned the atlases to the cortical surface. We computed head meshes (boundary element method [BEM] surfaces) using fieldtrip (*Oostenveld et al., 2011*) and the head shape model using MNE (*Gramfort et al., 2014*). Next, we created the transformation matrix by co-registering the headlock fiducials to the head model separately for each subject and session. A source space (4096 vertices per hemisphere, recursively subdivided octahedron) was computed for each hemisphere, surfaces were converted to a BEM and the BEM solution was computed using MNE. We baseline-corrected the stimulus and response epochs using a baseline interval from 0.35 to 0.1 s before stimulus onset and computed a data covariance matrix from the stimulus epochs separately for each subject and session. The leadfield (forward solution) was computed using the subject and session-specific transformation matrix, source space, and BEM solution. Finally, the LCMV spatial filters (*Van Veen et al., 1997*) were constructed for each vertex in each ROI from the forward solution and the data covariance matrix. As ROIs we focused on a number of topographically organized visual cortical field maps (*Wang et al., 2015*) and three regions exhibiting action-selective activity lateralization in functional MRI (*de Gee et al., 2017*): the hand area of primary motor cortex (M1), the junction of intraparietal sulcus/postcentral sulcus IPS/PostCes, and a part of anterior intraparietal sulcus (aIPS).

## Regions of interest

We delineated power at specific ROIs that have been shown to be involved in decision-making, the decision-related dynamics of which have been characterized in detail in previous work (*Wilming et al., 2020*; *Murphy et al., 2021*). During decision formation, sensory evidence is encoded in visual cortex. This signal is accumulated across time into a DV in association cortex and transformed into a motor action in motor cortex (*Gold and Shadlen, 2007*; *Wang, 2008*; *Siegel et al., 2011*). Specifically, we selected the ROIs from the Wang atlas (*Wang et al., 2015*) and combined them into the following clusters of interest: primary occipital cortex V1, early occipital cortex V2-4, dorsal occipital cortex V3A/B, intraparietal sulcus IPS0/1 and IPS2/3, lateral occipital cortex LO1 and LO2, temporal occipital area MT+ (MT and MST), ventral occipital cortex VO1 and VO2, parahippocampal cortex PCH1 and

PCH2. We used the following regions that have previously been identified to show choice-predictive lateralized activity (*de Gee et al., 2017*): anterior intraparietal sulcus aIPS, intraparietal sulcus/post-central sulcus IPS/PostCes, hand area of primary motor cortex M1.

## Assessment of spatial leakage between source estimates for neighboring regions

To assess the level of spatial leakage of our source estimates, we correlated the weights of the spatial filter used for beamforming (*Figure 2—figure supplement 1*). This correlation was computed separately for each subject, session, and hemisphere. To evaluate this correlation parametrically as a function of the spatial distance between sources (i.e., vertices), we averaged correlations across vertex pairs with the same distance, ranging from 0 to 5 cm in steps of 0.5 cm and finally collapsed across sessions, hemispheres, and subjects (*Figure 2—figure supplement 1A*). To obtain a matrix of correlations between all pairs of the ROIs shown in *Figure 2*, we first randomly sampled one vertex from a 'reference ROI' and correlated the spatial filters for this vertex with those of (randomly selected) vertices from all other ROIs of the set shown in *Figure 2*. We repeated this procedure several (*N*=30) times, averaged the resulting correlation coefficients across iterations, and then averaged further across sessions, hemispheres, and subjects. This yielded the correlations for the reference ROI with all the rest in the correlation matrix from *Figure 2—figure supplement 1B*. The procedure was then repeated for the next reference ROI until all cells of the lower triangular part of the matrix were filled with entries.

As highlighted in Results, our analyses yielded distinct, and physiologically plausible, functional profiles across areas. Such differences cannot be accounted for by leakage (*Figure 2*). Most importantly, our current analyses focus on the impact of history bias on the build-up of action-selective activity in downstream, action-related areas. We chose to focus on the M1 hand area in order to avoid hard-to-interpret comparisons between neighboring action-related regions. *Figure 2* is intended as a demonstration of the data quality (showing sensible signatures for all ROIs) and as a context for the interpretation of our main neural results from M1 shown in the subsequent figures.

## Definition of the time window of linear build-up of lateralized activity in M1

To test for a neural correlate of a bias in drift rate, we first determined the time window of the approximately linear build-up of lateralized activity in M1. During evidence accumulation, choice-predictive motor preparatory activity (a lateralized suppression of beta-band power) builds up contra- vs. ipsilateral to the upcoming button-press. This signal has been shown to exhibit the hallmark signatures of evidence accumulation postulated by the drift diffusion model (*Donner et al., 2009*; *de Lange et al., 2013*; *Pape and Siegel, 2016*). Hence, we used this signal as a neural correlate of the accumulated evidence. To determine the time window of evidence accumulation, we fitted a bilinear regression to the slope of the beta-band (12–36 Hz) power contra- vs. ipsilateral to the button-press in M1 pooled across environmental conditions and averaged across trials (*Figure 4A*). We used a time window with a buffer of 250 ms after the intersection point of the fitted lines and 50 ms before the minimum of the beta lateralization to test our hypotheses.

## Removal of beta rebound from previous trial

After the motor-response, beta lateralization flips its sign – the so-called beta rebound (*Pfurtscheller et al., 1996*). This signal leaks into the next trial, which may cause a motor-response alternation bias (*Pape and Siegel, 2016*). We computed the beta rebound as the beta-band time course contra- vs. ipsilateral to the previous button-press in M1, pooled across environmental conditions and averaged across trials, and normalized it to a unit vector $r$, separately for each subject. In control analyses for the results from *Figure 4B and C* we removed the beta rebound from the time course of the beta-band lateralization to isolate the effect of the bias adjustment to the statistical structure of the environment. The residual beta time course $y^*$ was computed as the difference of the original beta time course $y$ and its orthogonal projection with the beta rebound:

$$y^* = y - \left(y^T r\right) r. \tag{3}$$

## Assessment of bias-dependent dynamics of action-selective activity

To finally test for a bias-dependent evidence accumulation, we analyzed the beta lateralization conditioned on the behavioral bias at each single trial binned into three bins: a low bin corresponding to a bias for a 'down' choice, a medium bin with a bias close to zero, and a high bin with a bias for an 'up' choice (see section Single-trial bias estimates for details) (*Figure 1E*). The single-trial bias shifts the current choice at a given level of evidence. Consequently, the single-trial bias bins correlated with the final choice. The low bin primarily contained trials that resulted in a 'down' choice and the high bin primarily contained trials that resulted in an 'up' choice. To remove the effect of the final choice to isolate the effect of the single-trial bias, we subsampled the data such that each bias bin contained an equal number of up and down choices separately for each subject. To this end, we randomly drew the number of trials of the inferior choice from the data containing the predominant choice, separately for each bin. We repeated this procedure 1000 times and averaged the data across the draws. We finally computed the time course of the residual beta-band activity of the subsampled data contra- vs. ipsilateral to the button-press for the up choice. Averaging across the low bin with a sign flip and the high bin (without sign flip) yielded the beta lateralization contra- vs. ipsilateral to the button-press that was mapped onto the choice that was in line with the bias (*Figure 4B*). We then computed the slope of the build-up of the beta lateralization during the previously defined time window of linear build-up of lateralized activity in M1 (see Definition of the time window of linear build-up of lateralized activity in M1; *Figure 4A*) via linear regression (*Figure 4C*). The subsampling procedure did not change the distribution of coherences (see *Figure 4—figure supplement 1*).

## Single-trial regression of history bias and evidence on action-selective activity

We used a linear regression model to quantify the influence of the current sensory evidence (i.e., the signed motion coherence) as well as of the single-trial bias on the single-trial modulation of M1 power lateralization during each time point $t$:

$$beta\_lat_t = \beta_0 + \beta_1 * coh + \beta_2 * bias \tag{4}$$

where $beta\_lat_t$ was the beta-power lateralization relative to the hand coding up-choices in a given block during time point t, $coh$ was the signed motion coherence, and $bias$ was the single trial bias. The power values for each time point $t$, frequency $f$, and sensor $c$ were normalized and baseline-corrected via the decibel (dB) transform before computing the beta lateralization: $dB_{t,f,c} = 10 * log_{10} \left( power_{t,f,c} \, baseline_{f,c} \right)$, where $baseline_{f,c}$ was the trial-averaged power during the baseline interval (350–100 ms before onset of coherent motion). All regressors as well as power values were z-scored prior to the regression analysis. We expected a negative influence of the signed motion coherence as well as of the single-trial bias on the motor beta lateralization contra- vs. ipsilateral to the button-press for up responses (*Figure 4D and E*).

For the analysis of the influence of the single-trial bias on the baseline M1 beta lateralization (350–100 ms before evidence onset; *Figure 3C*), we used an analogous regression analysis but without using the signed motion coherence as a regressor because the onset of coherent motion started only after the baseline interval:

$$beta\_lat_{baseline} = \beta_0 + \beta_1 * bias \tag{5}$$

## Single-trial regression of history bias and evidence on slope of action-selective activity

We used the corresponding regression analysis for the slope of the motor beta lateralization separately for current up and down responses:

$$beta\_slope_t = \beta_0 + \beta_1 * coh + \beta_2 * bias \tag{6}$$

To this end we computed the slope of the M1 beta lateralization time course using a sliding window of 200 ms. The slope for each time window $t$ as well as the regressors were z-scored before computing the regression. We plotted the beta weights at the center of each 200 ms time window that was used to compute the slope of the beta lateralization (*Figure 4D and E* gray line).

## Statistical tests

We used parametric two-tailed t tests to test the effect of the previous stimulus category on the shift and the slope of the psychometric function in order to also provide Bayes factors ($Bf$) (**Rouder et al., 2009**; **Figure 1C**). $Bf_{10} < \frac{1}{3}$ corresponds to evidence in favor of the null hypothesis, $Bf_{10} > 3$ refers to evidence for the alternative hypothesis, and $Bf_{10} = 1$ corresponds to inconclusive evidence. We used Pearson correlation for computing the partial correlation between the bias adjustment and performance (**Figure 1F** and **Figure 1—figure supplement 3**) as well as for computing the correlation coefficients of the LCMV beamformer weights across ROIs (**Figure 2—figure supplement 1**). We used nonparametric permutation tests (**Efron and Tibshirani, 1998**) with $N = 10,000$ permutations to test the previous stimulus weights (**Figure 1D**), the baseline state of the beta lateralization (**Figure 3B**), the slope of the build-up of motor preparatory activity (**Figure 4C**), as well as for the regression of the single-trial history bias on action-selective activity during the baseline interval (**Figure 3C**). Cluster-based permutation tests were used for time-frequency responses (**Figures 2 and 3A**) and for time courses of beta-band power (**Figure 4**, **Figure 2—figure supplement 2**). We used circular statistics (Rayleigh's test) to test the clustering of vector angles between the origin and the weights from the neutral environment as well as between the weights from neutral and the weights from the repetitive or alternating environments, respectively (**Figure 1—figure supplement 2A**). To test the difference in mean directions of adjustment between the repetitive and the alternating environment, we used a Hotelling test (**van den Brink et al., 2014**; **Figure 1—figure supplement 2A**).

## Acknowledgements

We thank Niklas Wilming for discussion on MEG source reconstruction and Jaime de la Rocha, Anne Urai, Bharath Chandra Talluri, and Alessandro Toso for discussion and comments on the manuscript. We thank Alessandro Toso for help with assessment of leakage of MEG source estimates. Funding: This work has been supported by the Deutsche Forschungsgemeinschaft (DFG), projects DO1240-4-1, DO1240_2–2, and SFB 936 – 178316478 – Z3 and by the Federal Ministry of Education and Research (BMBF), project 01EW2007B (all to THD).

## Additional information

### Funding

| Funder | Grant reference number | Author |
| --- | --- | --- |
| Deutsche Forschungsgemeinschaft | projects DO1240-4-1 | Tobias H Donner |
| Bundesministerium für Bildung und Forschung | project 01EW2007B | Tobias H Donner |
| Sonderforschungsbereich (SFB) 936 | 178316478 - Z3 | Tobias H Donner |
| Deutsche Forschungsgemeinschaft | DO1240_2-2 | Tobias H Donner |

The funders had no role in study design, data collection and interpretation, or the decision to submit the work for publication.

### Author contributions

Anke Braun, Conceptualization, Data curation, Software, Formal analysis, Investigation, Visualization, Methodology, Writing – original draft, Writing – review and editing; Tobias H Donner, Conceptualization, Supervision, Funding acquisition, Writing – original draft, Project administration, Writing – review and editing

### Author ORCIDs

Anke Braun ⬤ https://orcid.org/0000-0002-1946-7765
Tobias H Donner ⬤ https://orcid.org/0000-0002-7559-6019

## Ethics

All participants gave their written informed consent. The experiment was approved by the local ethical review board (Ärztekammer Hamburg reference number PV4714).

Reviewer #1 (Public Review): https://doi.org/10.7554/eLife.86740.3.sa1
Reviewer #2 (Public Review): https://doi.org/10.7554/eLife.86740.3.sa2
Reviewer #3 (Public Review): https://doi.org/10.7554/eLife.86740.3.sa3
Author Response https://doi.org/10.7554/eLife.86740.3.sa4

# Additional files

## Supplementary files

• MDAR checklist

## Data availability

Raw MEG data are available at https://www.fdr.uni-hamburg.de/record/13475. Source reconstructed MEG data are available at https://www.fdr.uni-hamburg.de/record/13197. Behavioral data is available at https://www.fdr.uni-hamburg.de/record/13517. All data is available under Creative Commons Attribution 4.0 International License. The ethics protocol disallows sharing MRI data via a public repository. Data may be shared however within the context of a collaboration. No proposal is needed. In order to obtain the data, please email Anke Braun (anke.braun86@gmail.com) and Tobias H. Donner (t.donner@uke.de). The code and data immediately underlying all main and figure supplements are publicly available on https://github.com/DonnerLab/2023_BraunA_Adaptive_biasing_of_action-selective_cortical_build-up_activity_by_stimulus_history (copy archived at *DonnerLab, 2023a*) under GNU General Public LicenseVersion 2. The logistic regression model with history bias was fitted using a toolbox from *Fründ et al., 2014*, which is publicly available under https://bitbucket.org/mackelab/serial_decision/src/master/. Preprocessing of MEG data was done using a Fieldtrip pipeline from *Urai and Donner, 2022*, which is publicly available on https://github.com/DonnerLab/2022_Urai_choice-history_MEG (copy archived at *DonnerLab, 2022*). Source reconstruction of MEGdata was done using pymeg (*Wilming et al., 2020*), which is publicly available under https://github.com/DonnerLab/pymeg (copy archived at *DonnerLab, 2023b*).

The following datasets were generated:

| Author(s) | Year | Dataset title | Dataset URL | Database and Identifier |
|---|---|---|---|---|
| Braun A, Donner THD | 2023 | Source Reconstructed MEG Data for Adaptive biasing of action-selective cortical build-up activity by stimulus history | https://doi.org/10.25592/uhhfdm.13196 | Forschungsdatenrepositorium Uni Hamburg, 10.25592/uhhfdm.13196 |
| Braun A, Donner THD | 2023 | Raw MEG Data for Adaptive biasing of action-selective cortical build-up activity by stimulus history | https://doi.org/10.25592/uhhfdm.13474 | Forschungsdatenrepositorium Uni Hamburg, 10.25592/uhhfdm.13474 |
| Braun A, Donner THD | 2023 | Behavioral data for Adaptive biasing of action-selective cortical build-up activity by stimulus history | https://doi.org/10.25592/uhhfdm.13516 | Forschungsdatenrepositorium Uni Hamburg, 10.25592/uhhfdm.13516 |

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
