## [Editor Report · eLife assessment]

In uncertain conditions, decisions are not made in isolation but are rather biased by the recent past. This new work provides **valuable** insights into these history biases in human perceptual decision-making, by characterizing the neural correlates of stimulus history biases and their short-term dynamics. The study provides **compelling** behavioral and MEG evidence that humans adapt their history biases to the correlation structure of uncertain sensory environments.

---

## [Referee Report · Reviewer #1 (Public Review)]

This paper aims to study the effects of choice history on action-selective beta band signals in human MEG data during a sensory evidence accumulation task. It does so by placing participants in three different stochastic environments, where the outcome of each trial is either random, likely to repeat, or likely to alternate across trials. The authors provide good behavioural evidence that subjects have learnt these statistics (even though they are not explicitly told about them) and that they influence their decision-making, especially on the most difficult trials (low motion coherence). They then show that the primary effect of choice history on lateralised beta-band activity, which is well-established to be linked to evidence accumulation processes in decision-making, is on the slope of evidence accumulation rather than on the baseline level of lateralised beta.

The strengths of the paper are that it is: (i) very well analysed, with compelling evidence in support of its primary conclusions; (ii) a well-designed study, allowing the authors to investigate the effects of choice history in different stochastic environments.

There are no major weaknesses to the study. On the other hand, investigating the effects of choice/outcome history on evidence integration is a fairly well-established problem in the field. As such, I think that this provides a valuable contribution to the field, rather than being a landmark study that will transform our understanding of the problem.

The authors have achieved their primary aims and I think that the results support their main conclusions. One outstanding question in the analysis is the extent to which the source-reconstructed patches in Figure 2 are truly independent of one another (as often there is 'leakage' from one source location into another, and many of the different ROIs have quite similar overall patterns of synchronisation/desynchronisation.). A possible way to investigate this further would be to explore the correlation structure of the LCMV beamformer weights for these different patches, to ask how similar/dissimilar the spatial filters are for the different reconstructed patches.

The revised paper now states explicitly how source-reconstructed patches are indeed affected by leakage, but also why the focus of the authors on differences (rather than similarities) between patches leaves their findings and conclusions essentially unaffected by this intrinsic limitation of cortical source reconstruction from surface MEG data.

---

## [Referee Report · Reviewer #2 (Public Review)]

In this work, the authors use computational modeling and human neurophysiology (MEG) to uncover behavioral and neural signatures of choice history biases during sequential perceptual decision-making. In line with previous work, they see neural signatures reflecting choice planning during perceptual evidence accumulation in motor-related regions, and further show that the rate of accumulation responds to structured, predictable environments suggesting that statistical learning of environment structure in decision-making can adaptively bias the rate of perceptual evidence accumulation via neural signatures of action planning. The data and evidence show subtle but clear effects, and are consistent with a large body of work on decision-making and action planning.

Overall, the authors achieved what they set out to do in this nice study, and the results, while somewhat subtle in places, support the main conclusions. This work will have an impact within the fields of decision-making and motor planning, linking statistical learning of structured sequential effects in sense data to evidence accumulation and action planning.

Strengths:

- The study is elegantly designed, and the methods are clear and generally state-of-the-art

- The background leading up to the study is well described, and the study itself conjoins two bodies of work - the dynamics of action-planning processes during perceptual evidence accumulation, and the statistical learning of sequential structure in incoming sense data

- Careful analyses effectively deal with potential confounds (e.g., baseline beta biases)

Weaknesses (after revision):

- The treatment of "awareness" of task structure is left as a somewhat open, potentially important question.

---

## [Referee Report · Reviewer #3 (Public Review)]

This study examines how the correlation structure of a perceptual decision-making task influences history biases in responding. By manipulating whether stimuli were more likely to be repetitive or alternating, they found evidence from both behavior and a neural signal of decision formation that history biases are flexibly adapted to the environment. On the whole, these findings are supported across an impressive range of detailed behavioral and neural analyses. The methods and data from this study will likely be of interest to cognitive neuroscience and psychology researchers. The results provide new insights into the mechanisms of perceptual decision-making.

The behavioral analyses are thorough and convincing, supported by a large number of experimental trials (~600 in each of 3 environmental contexts) in 38 participants. The psychometric curves provide clear evidence of adaptive history biases. The paper then goes on to model the effect of history biases at the single trial level, using an elegant cross-validation approach to perform model selection and fitting. The results support the idea that, with trial-by-trial accuracy feedback, the participants adjusted their history biases due to the previous stimulus category, depending on the task structure in a way that contributed to performance.

The paper then examines MEG signatures of decision formation, to try to identify neural signatures of these adaptive biases. Looking specifically at motor beta lateralization, they found no evidence that starting-level bias due to the previous trial differed depending on the task context. This suggests that the adaptive bias unfolds in the dynamic part of the decision process, rather than reflecting a starting level bias. This is supported by analysis of lateralization relative to the chosen hand as a proxy for a decision variable (DV), whose slope is shown to be influenced by these adaptive biases.

---

## [Author Response]

The following is the authors’ response to the original reviews.

**eLife assessment**
This valuable work provides new insights into history-dependent biases in human perceptual decisionmaking. It provides compelling behavioral and MEG evidence that humans adapt their historydependent to the correlation structure of uncertain sensory environments. Further neural data analyses would strengthen some of the findings, and the studied bias would be more accurately framed as a stimulus- or outcome-history bias than a choice-history bias because tested subjects are biased not by their previous choice, but by the previous feedback (indicating the category of the previous stimulus).

Thank you for your constructive evaluation of our manuscript. We have followed your suggestion to frame the studied bias as ‘stimulus history bias’. We now use this term whenever referring to our current results. Please note that we instead use the generic term ‘history bias’ when referring to the history biases studied in the previous literature on this topic in general. This is because these biases were dependent on previous choice(s), previous stimuli, or previous outcomes, or combinations of some (or all) of these factors. We have also added several of your suggested neural data analyses so as to strengthen the support for our conclusions, and we have elaborated on the Introduction so as to clarify the gaps in the literature that our study aims to fill. Our revisions are detailed in our replies below. We also took the liberty to reply to some points in the Public Review, which we felt called for clarification of the main aims (and main contribution) of our study.

**Reviewer #1 (Public Review):**
This paper aims to study the effects of choice history on action-selective beta band signals in human MEG data during a sensory evidence accumulation task. It does so by placing participants in three different stochastic environments, where the outcome of each trial is either random, likely to repeat, or likely to alternate across trials. The authors provide good behavioural evidence that subjects have learnt these statistics (even though they are not explicitly told about them) and that they influence their decision-making, especially on the most difficult trials (low motion coherence). They then show that the primary effect of choice history on lateralised beta-band activity, which is well-established to be linked to evidence accumulation processes in decision-making, is on the slope of evidence accumulation rather than on the baseline level of lateralised beta.The strengths of the paper are that it is: (i) very well analysed, with compelling evidence in support of its primary conclusions; (ii) a well-designed study, allowing the authors to investigate the effects of choice history in different stochastic environments.

Thank you for pointing out these strengths of our study.

There are no major weaknesses to the study. On the other hand, investigating the effects of choice/outcome history on evidence integration is a fairly well-established problem in the field. As such, I think that this provides a valuable contribution to the field, rather than being a landmark study that will transform our understanding of the problem.

Your evaluation of the significance of our work made us realize that we may have failed to bring across the main gaps in the literature that our current study aimed to fill. We have now unpacked this in our revised Introduction.

Indeed, many previous studies have quantified history-dependent biases in perceptual choice. However, the vast majority of those studies used tasks without any correlation structure; only a handful of studies have quantified history biases in tasks entailing structured environments, as we have done here (Abrahamyan et al., 2016; Kim et al., 2017; Braun et al., 2018; Hermoso-Mendizabal et al., 2020). The focus on correlated environments matters from an ecological perspective, because (i) natural environments are commonly structured rather than random (a likely reason for history biases being so prevalent in the first place), and (ii) history biases that change flexibly with the environmental structure are a hallmark of adaptive behavior. Critically, the few previous studies that have used correlated environments and revealed flexible/adaptive history biases were purely behavioral. Ours is the first to characterize the neural correlates of adaptive history biases.

Furthermore, although several previous studies have identified neural correlates of history biases in standard perceptual choice tasks in unstructured environments (see (Talluri et al., 2021) for a brief overview), most have focused on static representations of the bias in ongoing activity preceding the new decision; only a single monkey physiology study has tested for both a static bias in the pre-stimulus activity and a dynamic bias building up during evidence accumulation (Mochol et al., 2021). Ours is the first demonstration of a dynamic bias during evidence accumulation in the human brain.

The authors have achieved their primary aims and I think that the results support their main conclusions. One outstanding question in the analysis is the extent to which the source-reconstructed patches in Figure 2 are truly independent of one another (as often there is 'leakage' from one source location into another, and many of the different ROIs have quite similar overall patterns of synchronisation/desynchronisation.).

We do not assume (and nowhere state) that the different ROIs are “truly independent” of one another. In fact, patterns of task-related power modulations of neural activity would be expected to be correlated between many visual and action-related cortical areas even without leakage (due to neural signal correlations). So, one should not assume independence even for intracortically recorded local field potential data, fMRI data, or other data with minimal spatial leakage effects. That said, we agree that filter leakage will add a (trivial) component to the similarity of power modulations across ROIs, which can and should be quantified with the analysis you propose.

A possible way to investigate this further would be to explore the correlation structure of the LCMV beamformer weights for these different patches, to ask how similar/dissimilar the spatial filters are for the different reconstructed patches.

Thank you for suggesting this analysis, which provides a very useful context for interpreting the pattern of results shown in our Figure 2. We have now computed (Pearson) correlation coefficients of the LCMV beamformer weights across the regions of interest. The results are shown in the new Figure 2 – figure supplement 1. This analysis provided evidence for minor leakage between the source estimates for neighboring cortical regions (filter correlations <= than 0.22 on average across subjects) and negligible leakage for more distant regions. We now clearly state this when referring to Figure 2.

That said, we would also like to clarify our reasoning behind Figure 2. Our common approach to these source-reconstructed MEG data is to focus on the differences, rather than the similarities between ROIs, because the differences cannot be accounted for by leakage. Our analyses show clearly distinct, and physiologically plausible functional profiles across ROIs (motion coherence encoding in visual regions, action choice coding in motor regions), in line with other work using our general approach (Wilming et al., 2020; Murphy et al., 2021; Urai and Donner, 2022).

Most importantly, our current analyses focus on the impact of history bias on the build-up of actionselective activity in downstream, action-related areas; and we chose to focus on M1 only in order to avoid hard-to-interpret comparisons between neighboring action-related regions. Figure 2 is intended as a demonstration of the data quality (showing sensible signatures for all ROIs) and as a context for the interpretation of our main neural results from M1 shown in the subsequent figures. So, all our main conclusions are unaffected by leakage between ROIs.

We have now clarified these points in the paper.

**Reviewer #2 (Public Review):**
In this work, the authors use computational modeling and human neurophysiology (MEG) to uncover behavioral and neural signatures of choice history biases during sequential perceptual decision-making. In line with previous work, they see neural signatures reflecting choice planning during perceptual evidence accumulation in motor-related regions, and further show that the rate of accumulation responds to structured, predictable environments suggesting that statistical learning of environment structure in decision-making can adaptively bias the rate of perceptual evidence accumulation via neural signatures of action planning. The data and evidence show subtle but clear effects, and are consistent with a large body of work on decision-making and action planning.Overall, the authors achieved what they set out to do in this nice study, and the results, while somewhat subtle in places, support the main conclusions. This work will have impact within the fields of decisionmaking and motor planning, linking statistical learning of structured sequential effects in sense data to evidence accumulation and action planning.Strengths:The study is elegantly designed, and the methods are clear and generally state-of-the-artThe background leading up to the study is well described, and the study itself conjoins two bodies of work - the dynamics of action-planning processes during perceptual evidence accumulation, and the statistical learning of sequential structure in incoming sense dataCareful analyses effectively deal with potential confounds (e.g., baseline beta biases)

Thank you for pointing out these strengths of our study.

Weaknesses:Much of the study is primarily a verification of what was expected based on previous behavioral work, with the main difference (if I'm not mistaken) being that subjects learn actual latent structure rather than expressing sequential biases in uniform random environments.

As we have stated in our reply to the overall assessment above, we realize that we may have failed to clearly communicate the novelty of our current results, and we have revised our Introduction accordingly. It is true that most previous studies of history biases in perceptual choice have used standard tasks without across-trial correlation structure. Only a handful of studies have quantified history biases in tasks entailing structured environments that varied from one condition to the next (Abrahamyan et al., 2016; Kim et al., 2017; Braun et al., 2018; Hermoso-Mendizabal et al., 2020), and showed that history biases change flexibly with the environmental structure. Our current work adds to this emerging picture, using a specific task setting analogous to one of these previous studies done in rats (Hermoso-Mendizabal et al., 2020).

Critically, all the previous studies that have revealed flexible/adaptive history biases in correlated environments were purely behavioral. Ours is the first to characterize the neural correlates of adaptive history biases. And it is also the very first demonstration of a dynamic history-dependent bias (i.e., one that gradually builds up during evidence accumulation) in the human brain.

Whether this difference - between learning true structure or superstitiously applying it when it's not there - is significant at the behavioral or neural level is unclear. Did the authors have a hypothesis about this distinction? If the distinction is not relevant, is the main contribution here the neural effect?

We are not quite sure what exactly you mean with “is significant”, so we will reply to two possible interpretations of this statement.

The first is that you may be asking for evidence for any difference between the estimated history biases in the structured (i.e., Repetitive, Alternating) vs. the unstructured (i.e., Neutral) environments used in our experiment. We do, in fact, provide quantitative comparisons between the history biases in the structured and Neutral environments at the behavioral level. Figure 1D and Figure 1 – figure supplement 2A and accompanying text show a robust and statistically significant difference in history biases. Specifically, the previous stimulus weights differ between each of the biased environments and the Neutral environment and the weights shifted in expected and opposite directions for both structured environments, indicating a tendency to repeat the previous stimulus category in Repetitive and vice versa in Alternating (Figure1D). Going further, we also demonstrate that the adjustment of the history is behaviorally relevant in that it improves performance in the two structured environments, but not in the unstructured environment (Figure 1F and Figure 1 – figure supplement 2A and figure supplement 3).

The second is that you refer to the question of whether the history biases are generated via different computations in structured vs. random environments. Indeed, this is a very interesting and important question. We cannot answer this question based on the available results, because we here used a statistical (i.e., descriptive) model. Addressing this question would require developing and fitting a generative model of the history bias and comparing the inferred latent learning processes between environments. This is something we are doing in ongoing work.

The key effects (Figure 4) are among the more statistically on-the-cusp effects in the paper, and the Alternating group in 4C did not reliably go in the expected direction. This is not a huge problem per se, but does make the key result seem less reliable given the clear reliability of the behavioral results

The model-free analyses in Figure 3C and 4B, C from the original version of our manuscript were never intended to demonstrate the “key effects”, but only as supplementary to the results from the modelbased analyses in Figures 3C and 4D, E in our current version of the manuscript. The latter show the “key effects” because they are a direct demonstration of the shaping of build-up of action-selective activity by history bias.

To clarify this, we now decided to focus Figures 3 and 4 on the model-based analyses only. This decision was further supported by noticing a confound in our model-independent analyses in new control analyses prompted by Reviewer #3.

Please note that the alternating bias in the Alternating environment is also less strong at the behavioral level compared to the bias in the Repetitive condition (see Figure 1D). A possible explanation is that a sequence of repetitive stimuli produces stronger prior expectations (for repetition) than an equally long sequence of alternating stimuli (Meyniel et al., 2016). This might also induce the bias to repeat the previous stimulus category in the Neutral condition (Figure 1D). Moreover, this intrinsic repetition bias might counteract the bias to alternate the previous stimulus category in Alternating.

The treatment of "awareness" of task structure in the study (via informal interviews in only a subsample of subjects) is wanting

Agreed. We have now removed this statement from Discussion.

**Reviewer #3 (Public Review):**
This study examines how the correlation structure of a perceptual decision making task influences history biases in responding. By manipulating whether stimuli were more likely to be repetitive or alternating, they found evidence from both behavior and a neural signal of decision formation that history biases are flexibly adapted to the environment. On the whole, these findings are supported across an impressive range of detailed behavioral and neural analyses. The methods and data from this study will likely be of interest to cognitive neuroscience and psychology researchers. The results provide new insights into the mechanisms of perceptual decision making.The behavioral analyses are thorough and convincing, supported by a large number of experimental trials (~600 in each of 3 environmental contexts) in 38 participants. The psychometric curves provide clear evidence of adaptive history biases. The paper then goes on to model the effect of history biases at the single trial level, using an elegant cross-validation approach to perform model selection and fitting. The results support the idea that, with trial-by-trial accuracy feedback, the participants adjusted their history biases due to the previous stimulus category, depending on the task structure in a way that contributed to performance.

Thank you for these nice words on our work.

The paper then examines MEG signatures of decision formation, to try to identify neural signatures of these adaptive biases. Looking specifically at motor beta lateralization, they found no evidence that starting-level bias due to the previous trial differed depending on the task context. This suggests that the adaptive bias unfolds in the dynamic part of the decision process, rather than reflecting a starting level bias. The paper goes on to look at lateralization relative to the chosen hand as a proxy for a decision variable (DV), whose slope is shown to be influenced by these adaptive biases.This analysis of the buildup of action-selective motor cortical activity would be easier to interpret if its connection with the DV was more explicitly stated. The motor beta is lateralized relative to the chosen hand, as opposed to the correct response which might often be the case. It is therefore not obvious how the DV behaves in correct and error trials, which are combined together here for many of the analyses.

We have now unpacked the connection of the action-selective motor cortical activity and decision variable in the manuscript, as follows:

“This signal, referred to as ‘motor beta lateralization’ in the following, has been shown to exhibit hallmark signatures of the DV, specifically: (i) selectivity for choice and (ii) ramping slope that depends on evidence strength (Siegel et al., 2011; Murphy et al., 2021; O’Connell and Kelly, 2021).”

Furthermore, we have added a figure of the time course of the motor beta lateralization separately for correct and error trials, locked to both stimulus onset and to motor response (Figure 2 – figure supplement 2). This signal reached statistical significance earlier for correct than error trials, and during the stimulus interval it ramped to a larger (i.e., more negative) amplitude for correct trials (Figure 2 – figure supplement 2, left). But the signal was indistinguishable in amplitude between correct and error trials around the time of the motor response (Figure 2 – figure supplement 2, right). This pattern matches what would be expected for a neural signature of the DV, because errors are more frequently made on weak-evidence trials than correct choices and because even for matched evidence strength, the DV builds up more slowly before error trials in accumulator models (Ratcliff and McKoon, 2008).

--

As you will see, all three reviewers found your work to provide valuable insights into history-dependent biases during perceptual decision-making. During consultation between reviewers, there was agreement that what is referred as a choice-history bias in the current version of the manuscript should rather be framed as a stimulus- or outcome-history bias (despite the dominant use of the term 'choicehistory' bias in the existing literature), and the reviewers pointed toward further analyses of the neural data which they thought would strengthen some of the claims made in the preprint. We hope that these comments will be useful if you wish to revise your preprint.

We are pleased to hear that the reviewers think our work provides valuable insights into historydependent biases in perceptual decision-making. We thank you for your thoughtful and constructive evaluation of our manuscript.

We have followed your suggestion to frame the studied bias as ‘stimulus history bias’. We now use this term whenever referring to our current results. Please note that we instead use the generic term ‘history bias’ when referring to the history biases studied in the previous literature on this topic in general. This is because these biases were dependent on previous choice(s), previous stimuli, or previous outcomes, or combinations of some (or all) of these factors.

We have also performed several of your suggested neural data analyses so as to strengthen the support for our conclusions.

**Reviewer #1 (Recommendations For The Authors):**
One suggestion is to explore the correlation structure of the LCMV beam former weights for the regions of interest in the study, for the reasons outlined in my public review.

Again, thank you for suggesting this analysis, which provides a very useful context for interpreting the pattern of results shown in our Figure 2. We have now computed (Pearson) correlation coefficients of the LCMV beamformer weights across the regions of interest. The results are shown in the new Figure 2 – figure supplement 1. This analysis provided evidence for minor leakage between the source estimates for neighboring cortical regions (filter correlations <= than 0.22 on average across subjects) and negligible leakage for more distant regions. We now clearly state this when referring to Figure 2.

That said, we would also like to clarify our reasoning behind Figure 2. Our common approach to these source-reconstructed MEG data is to focus on the differences, rather than the similarities between ROIs, because the differences cannot be accounted for by leakage. Our analyses show clearly distinct, and physiologically plausible functional profiles across ROIs (motion coherence encoding in visual regions, action choice coding in motor regions), in line with other work using our general approach (Wilming et al., 2020; Murphy et al., 2021; Urai and Donner, 2022).

Most importantly, our current analyses focus on the impact of history bias on the build-up of actionselective activity in downstream, action-related areas; and we chose to focus on M1 only in order to avoid hard-to-interpret comparisons between neighboring action-related regions. Figure 2 is intended as a demonstration of the data quality (showing sensible signatures for all ROIs) and as a context for the interpretation of our main neural results from M1 shown in the subsequent figures. So, all our main conclusions are unaffected by leakage between ROIs.

We have now clarified also these points in the paper.

I also wondered if the authors had considered:(i) the extent to which the bias changes across time, as the transition probabilities are being learnt across the experiment? given that these are not being explicitly instructed to participants, is any modelling possible of how the transition structure is itself being learnt over time, and whether this makes predictions of either behaviour or neural signals?

We refer to this point in the discussion. The learning of the transition probabilities which can and should be addressed. This requires generative models that capture the learning of the transition structure over time (Yu and Cohen, 2009; Meyniel et al., 2016; Glaze et al., 2018; Hermoso-Mendizabal et al., 2020).

The fact that our current statistical modeling approach successfully captures the bias adjustment between environments implies that the learning must be sufficiently fast. Tracking this process explicitly would be an exciting and important endeavor for the future. We think it is beyond the scope of the present study focusing on the trial-by-trial effect of history bias (however generated) on the build-up of action-selective activity.

(ii) neural responses at the time of choice outcome - given that so much of the paper is about the update of information in different statistical environments, it seems a shame that no analyses are included of feedback processing, how this differs across the different environments, and how might be linked to behavioural changes at the next trial.

We agree that the neural responses to feedback are a very interesting topic. We currently analyze these in another ongoing project on (outcome) history bias in a foraging task. We will consider re-analyzing the feedback component in the current data set, in this new study as well.

However, this is distinct from the main question that is in the focus of our current paper – which, as elaborated above, is important to answer: whether and how adaptive history biases shape the dynamics of action-selective cortical activity in the human brain. While interesting and important, neural responses to feedback were not part of this question. So, we prefer to keep the focus of our paper on our original question.

**Reviewer #2 (Recommendations For The Authors):**
Minor:-pg. 7: "inconstant"-some citations (e.g., Barbosa 2020) are missing from the bibliography

Thank you for pointing this out. We have fixed these.

-figure S2 is very useful! could probably go in main text.

We agree that this figure is important. But we decided to show it in the Supplement (now Figure 1 – figure supplement 2) after careful consideration for two reasons. First, we wanted to put the reader’s focus on the stimulus weights, because it is those weights, which are flexibly adjusted to the statistics of the environment rather than the choice weights, which seem less adaptive (i.e., stereotypical across environments) and idiosyncratic. Second, plotting the previous stimulus weights only enabled to add the individual weights in the Neutral condition, which would have been to cluttered to add to figure S2.

For these reasons, we feel that this Figure is more suitable for expert readers with a special interest in the details of the behavioral analyses and would be better placed in the Supplement. These readers will certainly be able to find and interpret that information in the Supplement.

**Reviewer #3 (Recommendations For The Authors):**
I would suggest that a more in depth description of the previous literature that explains exactly how the features of the lateralized beta--as it is formulated here-- reflect the decision variable would assist with the readers' understanding. A demonstration of how the lateralized beta behaves under different coherence conditions, or for corrects vs errors, for example, might be helpful for readers.

We now provide a more detailed description of how/why the motor beta lateralization is a valid proxy of DV in the revised paper.

We have demonstrated the dependence of the ramping of the motor beta lateralization on the motion coherence using a regression model with current signed motion coherence as well as single trial bias as regressors. The beta weights describing the impact of the signed motion coherence on the amplitude as well as on the slope of the motor beta lateralization are shown in Figure 4G (now 4E). As expected, stronger motion coherence induces a steeper downward slope of the motor beta lateralization.

Furthermore, we have added a figure of the time course of the motor beta lateralization separately for correct and error trials, locked to both stimulus onset and to motor response (Figure 2 – figure supplement 2). This signal reached statistical significance earlier for correct than error trials, and during the stimulus interval it ramped to a larger (i.e., more negative) amplitude for correct trials (Figure 2 – figure supplement 2, left). But the signal was indistinguishable in amplitude between correct and error trials around the time of the motor response (Figure 2 – figure supplement 2, right).This pattern matches what would be expected for a neural signature DV, because errors are more frequently made on weakevidence trials than correct choices and because even for matched evidence strength, the DV builds up more slowly before error trials in accumulator models (Ratcliff and McKoon, 2008).

Finally, please note that our previous studies have demonstrated that the time course of the beta lateralization during the trial closely tracks the time course of a normative model-derived DV (Murphy et al., 2021) and that the motor beta ramping slope is parametrically modulated by motion coherence (de Lange et al., 2013), which is perfectly in line with the current results.

Along similar lines, around figures 3c and 4B, some control analyses may be helpful to clarify whether there are differences between the groups of responses consistent and inconsistent with the previous trial (e.g. correctness, coherence) that differ between environments, and also could influence the lateralized beta.

Thank you for pointing us to this important control analysis. We have done this, and indeed, it identified accuracy and motion strength as possible confounds (Author response image 1). Specifically, proportion correct as well as motion coherence were larger for consistent vs. inconsistent conditions in Repetitive and vice versa in Alternating. Those differences in accuracy and coherence might indeed influence the slope of the motor beta lateralization that our model-free analysis had identified, rendering the resulting difference between consistent and inconsistent difficult to interpret unambiguously in terms of bias. Thus, we have decided to drop the consistency (i.e., model-independent) analysis and focus completely on the modelbased analyses.

**Author response image 1. sa4fig1:** Proportion correct and motion coherence split by environment and consistency of current choice and previous stimulus. In the Repetitive environment (Rep.), accuracy and motion coherence are larger for current choice consistent vs. inconsistent with previous stimulus category and vice versa in the Alternating environment (Alt.).

Importantly, this decision has no implications for the conclusions of our paper: The model-independent analyses in the original versions of Figure 3 and 4 were only intended as a supplement to the most conclusive and readily interpretable results from the model-based analyses (now in Figs. 3C and 4D, E). The latter are the most direct demonstration of a shaping of build-up of action-selective activity by history bias, and they are unaffected by these confounds.

In addition, I wondered whether the bin subsampling procedure to match trial numbers for choice might result in unbalanced coherences between the up and down choices.

The subsampling itself did not cause any unbalanced coherences between the up and down choices, which we now show in Figure 4 – figure supplement 1. There was only a slight imbalance in coherences between up and down choices before the subsampling which then translated into the subsampled trials but the coherences were equally distributed before as compared to after the subsampling.

Also, please note that the purpose of this analysis was to make the neural bias directly “visible” in the beta lateralization data, rather than just regression weights. The issue does not pertain to the critical single-trial regression analysis, which yielded consistent results.

References

Abrahamyan A, Silva LL, Dakin SC, Carandini M, Gardner JL (2016) Adaptable history biases in human perceptual decisions. Proceedings of the National Academy of Sciences 113:E3548–E3557.

Braun A, Urai AE, Donner TH (2018) Adaptive History Biases Result from Confidence-weighted Accumulation of Past Choices. The Journal of Neuroscience:2189–17. de Lange FP, Rahnev DA,Donner TH, Lau H (2013) Prestimulus Oscillatory Activity over Motor Cortex Reflects Perceptual Expectations. Journal of Neuroscience 33:1400–1410.

Glaze CM, Filipowicz ALS, Kable JW, Balasubramanian V, Gold JI (2018) A bias–variance trade-off governs individual differences in on-line learning in an unpredictable environment. Nat Hum Behav 2:213–224.

Hermoso-Mendizabal A, Hyafil A, Rueda-Orozco PE, Jaramillo S, Robbe D, de la Rocha J (2020) Response outcomes gate the impact of expectations on perceptual decisions. Nat Commun 11:1057.

Kim TD, Kabir M, Gold JI (2017) Coupled Decision Processes Update and Maintain Saccadic Priors in a Dynamic Environment. The Journal of Neuroscience 37:3632–3645.

Meyniel F, Maheu M, Dehaene S (2016) Human Inferences about Sequences: A Minimal Transition Probability Model Gershman SJ, ed. PLOS Computational Biology 12:e1005260.

Mochol G, Kiani R, Moreno-Bote R (2021) Prefrontal cortex represents heuristics that shape choice bias and its integration into future behavior. Current Biology 31:1234-1244.e6.

Murphy PR, Wilming N, Hernandez-Bocanegra DC, Prat-Ortega G, Donner TH (2021) Adaptive circuit dynamics across human cortex during evidence accumulation in changing environments. Nat Neurosci 24:987–997.

O’Connell RG, Kelly SP (2021) Neurophysiology of Human Perceptual Decision-Making. Annu Rev Neurosci 44:495–516.

Ratcliff R, McKoon G (2008) The Diffusion Decision Model: Theory and Data for Two-Choice Decision Tasks. Neural Computation 20:873–922.

Siegel M, Engel AK, Donner TH (2011) Cortical Network Dynamics of Perceptual Decision-Making in the Human Brain. Frontiers in Human Neuroscience 5 Available at: http://journal.frontiersin.org/article/10.3389/fnhum.2011.00021/abstract [Accessed April 8, 2017].

Talluri BC, Braun A, Donner TH (2021) Decision making: How the past guides the future in frontal cortex. Current Biology 31:R303–R306.

Urai AE, Donner TH (2022) Persistent activity in human parietal cortex mediates perceptual choice repetition bias. Nat Commun 13:6015.

Wilming N, Murphy PR, Meyniel F, Donner TH (2020) Large-scale dynamics of perceptual decision information across human cortex. Nat Commun 11:5109.

Yu A, Cohen JD (2009) Sequential effects: Superstition or rational behavior. Advances in neural information processing systems 21:1873–1880.